# Variations in microbiota composition of laboratory mice influence *Citrobacter rodentium* infection via variable short-chain fatty acid production

**Lisa Osbelt**[1,2], **Sophie Thiemann**[1], **Nathiana Smit**[1], **Till Robin Lesker**[1], **Madita Schröter**[1], **Eric J. C. Gálvez**[1,3], **Kerstin Schmidt-Hohagen**[4], **Marina C. Pils**[5], **Sabrina Mühlen**[6,7], **Petra Dersch**[6,7], **Karsten Hiller**[4,8], **Dirk Schlüter**[2,9], **Meina Neumann-Schaal**[10], **Till Strowig**[1,3,11]*

1 Department of Microbial Immune Regulation, Helmholtz Centre for Infection Research, Braunschweig, Germany, 2 ESF International Graduate School on Analysis, Imaging and Modelling of Neuronal and Inflammatory Processes, Otto-von-Guericke University, Magdeburg, Germany, 3 Hannover Medical School, Hannover, Germany, 4 Department of Bioinformatics & Biochemistry, Technische Universität Braunschweig, Germany, 5 Mouse Pathology, Helmholtz Centre for Infection Research, Braunschweig, Germany, 6 Department of Molecular Infection Biology, Helmholtz Centre for Infection Research, Braunschweig, Germany, 7 Institute for Infectiology, University of Münster, Münster, Germany, 8 Computational Biology of Infection Research, Helmholtz Centre for Infection Research, Braunschweig, Germany, 9 Department of Medical Microbiology and Hospital Epidemiology, Hannover Medical School, Hannover, Germany, 10 Bacterial Metabolomics, Leibniz Institute DSMZ—German Collection of Microorganisms and Cell Cultures, Braunschweig, Germany, 11 Centre for Individualized Infection Medicine, Hannover, Germany

* till.strowig@helmholtz-hzi.de

**Data Availability Statement:** 16S rRNA gene sequencing data have been deposited in the NCBI

## Abstract

The composition of the intestinal microbiota influences the outcome of enteric infections in human and mice. However, the role of specific members and their metabolites contributing to disease severity is largely unknown. Using isogenic mouse lines harboring distinct microbiota communities, we observed highly variable disease kinetics of enteric *Citrobacter rodentium* colonization after infection. Transfer of communities from susceptible and resistant mice into germ-free mice verified that the varying susceptibilities are determined by microbiota composition. The strongest differences in colonization were observed in the cecum and could be maintained *in vitro* by coculturing cecal bacteria with *C. rodentium*. Cohousing of animals as well as the transfer of cultivable bacteria from resistant to susceptible mice led to variable outcomes in the recipient mice. Microbiome analysis revealed that a higher abundance of butyrate-producing bacteria was associated with the resistant phenotype. Quantification of short-chain fatty acid (SCFA) levels before and after infection revealed increased concentrations of acetate, butyrate and propionate in mice with delayed colonization. Addition of physiological concentrations of butyrate, but not of acetate and/or propionate strongly impaired growth of *C. rodentium in vitro*. *In vivo* supplementation of susceptible, antibiotic-treated and germ-free mice with butyrate led to the same level of protection, notably only when cecal butyrate concentration reached a concentration higher than 50 nmol/mg indicating a critical threshold for protection. In the recent years, commensal-derived primary and secondary bacterial metabolites emerged as potent modulators of

(Bioproject Database) under the accession number: PRJNA572605.

**Funding:** LO was funded by a stipend of the Europäischen Strukturfonds Sachsen Anhalt (ESF) and the international graduate school ABINEP (project number 44 100 32 030 ZS/2016/08/80645). The study is partly funded by the Helmholtz Centre for Infection Research. The funders had no role in study design, data collection and analysis, decision to publish, or preparation of the manuscript.

**Competing interests:** The authors have declared that no competing interests exist.

hosts susceptibility to infection. Our results provide evidence that variations in SCFA production in mice fed fibre-rich chow-based diets modulate susceptibility to colonization with *Enterobacteriaceae* not only in antibiotic-disturbed ecosystems but even in undisturbed microbial communities. These findings emphasise the need for microbiota normalization across laboratory mouse lines for infection experiments with the model-pathogen *C. rodentium* independent of investigations of diet and antibiotic usage.

## Author summary

The distinct composition of the gut microbiota in each individual results in variable metabolic activity and output of these communities, which influences the host, including resistance to enteric pathogens. Lack of reproducibility in biomedical research is nowadays frequently attributed to the microbiota, but little is known about which specific members and metabolites contribute to disease severity. Here, we use genetically identical mouse lines with variable microbiota compositions on a standardized diet and observed highly variable colonization with the enteric pathogen *Citrobacter rodentium* without antibiotics intervention. We found the same differences in formerly germ-free animals harbouring the respective donors microbiota and also *in vitro* by coculturing cecal bacteria from resistant and susceptible animals with *C. rodentium* showing that the phenotype is fully dependent on differences in the microbiota. We analysed the microbiome composition and found a higher abundance of butyrate-producing bacteria as well as increased levels of butyrate in resistant mice. By supplementation of susceptible and germ-free animals with butyrate, we could significantly lower the levels of colonization highlighting that commensal-derived primary and secondary bacterial metabolites are highly variable between laboratory animals from different vendors and are potent modulators of hosts susceptibility to infection with *C. rodentium*.

## Introduction

The gut microbiota, complex and dense microbial communities inhabiting the intestinal tract, has a large impact on the host immune system and decisive functions during infections. An important role of the microbiota is the prevention of pathogen colonization and host entry via direct and indirect microbiota-pathogen interactions, collectively termed colonization resistance [1]. These interactions include competition for essential nutrients and environmental niches or production of toxic and inhibitory compounds, but also immune-mediated mechanisms like proper induction of immune cells or enhancement of antibacterial pathways [2]. Disruption of the homeostatic microbiota caused by environmental factors such as antibiotic treatment or changes in diet can alter the initial composition and the amount of produced metabolites including short-chain fatty acids (SCFA) thereby altering the susceptibility towards enteric pathogens [3–6]. Furthermore, inter-individual differences in the microbiota composition have been associated with the susceptibility to enteric infections [7–9].

Enterohemorrhagic *Escherichia coli* (EHEC), enteropathogenic *E. coli* (EPEC) and *Citrobacter rodentium* are members of the family of gram-negative *Enterobacteriaceae* and belong to the family of attaching and effacing (A/E) lesion-forming bacteria. Importantly, EHEC and EPEC can cause severe intestinal inflammation and diarrhea. In addition EHEC strains expressing the highly potent Shiga toxin (Stx) cause nephrotoxicity resulting in severe cases in the death of infected individuals [10]. Since human EHEC and EPEC only induce modest

pathogenicity in antibiotic treated adult mice, *C. rodentium* is frequently used to mimic these infections in mice [10–12]. Several studies have indicated that alterations in the composition and function of the microbiota of mice impact the intestinal colonization with *C. rodentium*. For instance, treatment with certain antibiotics such as metronidazole has been shown to increase susceptibility by eradication of specific microbes, whereas other antibiotics such as streptomycin did not affect severity of infection indicating that specific, but yet unknown, bacteria confer resistance to *C. rodentium* [13]. Furthermore, transfer of fecal microbiota from resistant to susceptible mice could delay colonization and reduce mortality of the susceptible mice indicating the importance of the microbiota to protect against *C. rodentium* [14]. Upon fecal transplantation increased levels of interleukin (IL)-22 coupled with augmented antimicrobial peptides regenerating islet-derived 3 (Reg3)γ and Reg3β were measured showing that yet unknown bacteria are able to promote resistance via indirect, immune-mediated pathways [14]. In addition, colonization with a single kind of bacterium, i.e. segmented filamentous bacteria (SFB) reduces *C. rodentium* colonization of the epithelium but not the lumen via increased expression of genes associated with inflammation and anti-microbial defenses [15]. Moreover, higher ratios of Clostridia species were considered to influence the epithelial barrier, specifically mucus secretion, therefore, indirectly affecting luminal colonization of *C. rodentium*[16]. Within the lumen, the expansion of commensal *E. coli* has been shown to inhibit *C. rodentium* colonization by competition for monosaccharides as nutrient [17].

Genetically identical mouse lines with diverse microbiota compositions were utilized in numerous studies to identify novel mechanisms how microbiota composition modulates several disease settings e.g. DSS-induced colitis [18], *Salmonella* Typhimurium[19–20], malaria [21] and stroke [22]. Hence, we argued that these types of mouse lines could allow broadening the knowledge, which specific species and pathogen-commensal interactions contribute to protection against *C. rodentium*.

In the present study, we demonstrate that isogenic C57BL/6N mouse lines from different breeding facilities feature distinct microbiota profiles and significantly differ in their susceptibility to *C. rodentium* infection. We demonstrate that colonization kinetics are dependent on microbiota composition and could identify that resistant mice harbor a higher ratio and diversity of bacteria within the Firmicutes phylum, which lead to increased levels of short-chain fatty acids, especially butyrate directly inhibiting the growth of *C. rodentium*. Strikingly, at physiological concentrations butyrate was the most effective SCFA in limiting the growth of *C. rodentium* and was even more effective in combination with acetate or propionate to abrogate *C. rodentium* growth completely *in vitro*. Supplementation of butyrate in the drinking water lead to significantly lower colonization of susceptible and germ-free mice *in vivo*, indicating that a higher portion of butyrate and SCFA producers plays a key role in protection against *C. rodentium* infection. Therefore, naturally occurring variations in the microbiota of antibiotics-naive mice determine the amount of SCFAs present in the lumen and might serve as a marker species to predict the individual's susceptibility towards colonization with *Enterobacteriaceae*.

## Results

### Isogenic mouse lines feature distinct microbiota compositions and varying disease kinetics after *Citrobacter rodentium* infection

To quantify the influence of variations in microbiota composition in laboratory mice on susceptibility to *C. rodentium* infection independent of variation in diet and prior exposure to antibiotics, isogenic mouse lines were utilized. These mouse lines were on a highly related genetic background (C57Bl6/N, sublines Crl, Tac, Hsd and RJ) and were housed in the same animal facility under specific pathogen free hygienic conditions, and fed the same diet to reduce

the impact of experimental variables except microbiota composition. Characterization of microbiota composition using 16S rRNA gene sequencing revealed in line with previous studies substantial differences in microbial compositions in terms of species variability, species richness and complexity in each of the seven mouse lines tested (SPF-1—SPF-7) (Fig 1A and S1 Fig).

Age- and gender-matched SPF-1 –SPF-7 mice were then infected with $10^8$ CFU *C. rodentium* [23] by oral gavage (Fig 1B). Kinetics of *C. rodentium* colonization was noted to be significantly different between the different mouse lines (Fig 1C and 1D). In the first days after infection (day 1 and 3 post infection (p.i.)), SPF-1 mice showed the highest pathogen burden in stool, whereas SPF-2, SPF-3, and SPF-7 mice had the lowest colonization. (Fig 1C and 1D). Of note, SPF-2 and SPF-3 mice shared high similarities with regard to the microbiota composition, which translated into similar disease kinetics (Fig 1C and 1D, S1A–S1D Fig). Minor body weight loss was observed in all animals (< 5% of weight loss) with no significant differences in weight loss between mouse lines (S1E Fig).

Since SPF-1 (SPF-S) and SPF-2 (SPF-R) mice showed the highest difference in disease kinetics, we decided to further focus on these two mouse lines. To exclude that differences in the genotype between C57BL6/N sub-lines are responsible for the observed phenotype, SPF-S (susceptible) and SPF-R (resistant) mice were separately cohoused with germ-free (GF) C57BL6/NTac mice for four weeks, before all animals were infected with $10^8$ CFU *C. rodentium* by oral gavage (Fig 1E). Strikingly, exGF mice show the same difference in *C. rodentium* colonization as the respective SPF-S or SPF-R donor mice (Fig 1G). Analysis of microbiota composition after cohousing by 16S rRNA gene sequencing demonstrated that recipient exGF mice feature a highly similar microbiota composition in feces as respective donor mice (SPF-S, SPF-R) (Fig 1F). The results strongly support the conclusion that the observed phenotype depends on variations in microbiota composition.

## Variations in microbiota composition are correlated with lethal outcome of infections with Shiga toxin expressing *C. rodentium* strains

Shiga toxin (Stx) is a virulence factor of EHEC contributing to its strong pathogenicity. Recombinant expression of Shiga toxin in *C. rodentium* has been demonstrated to cause lethal infection accompanied by intestinal inflammation and kidney damage [12]. To test whether microbiota-dependent differences in *C. rodentium* colonization would also affect infection with a more pathogenic *C. rodentium* strain, SPF-S and SPF-R mice were infected with a Stx-producing *C. rodentium* strain [12] and fecal colonization as well as body weight loss and survival were analyzed. In line with the results using the less-pathogenic *C. rodentium* strain, the Stx-expressing *C. rodentium* strain also displayed delayed colonization in SPF-R mice during the first days of infection (Fig 2A). As likely direct consequence of higher colonization, SPF-S mice started to lose weight rapidly already after 3 days of infection, whereas SPF-R mice began to lose weight only at day 10 post infection (Fig 2B). Moreover, SPF-R mice displayed prolonged survival as SPF-S mice had a mean survival of 7 days, whereas SPF-R mice survived longer with a median survival of 11.5 days (Fig 2C). These findings demonstrate that delayed colonization of SPF-R mice lead to reduced body weight loss and prolonged survival even when infected with a lethal *C. rodentium* strain.

## Delayed colonization in SPF-R mice is visible in all intestinal organs and tissues but most prominent in the cecum

To study the infection process further, the characterization of the infection dynamic was extended. First, viable fecal CFUs were determined six hours after infection to quantify whether similar numbers of viable *C. rodentium* pass the gastrointestinal tract of both mouse

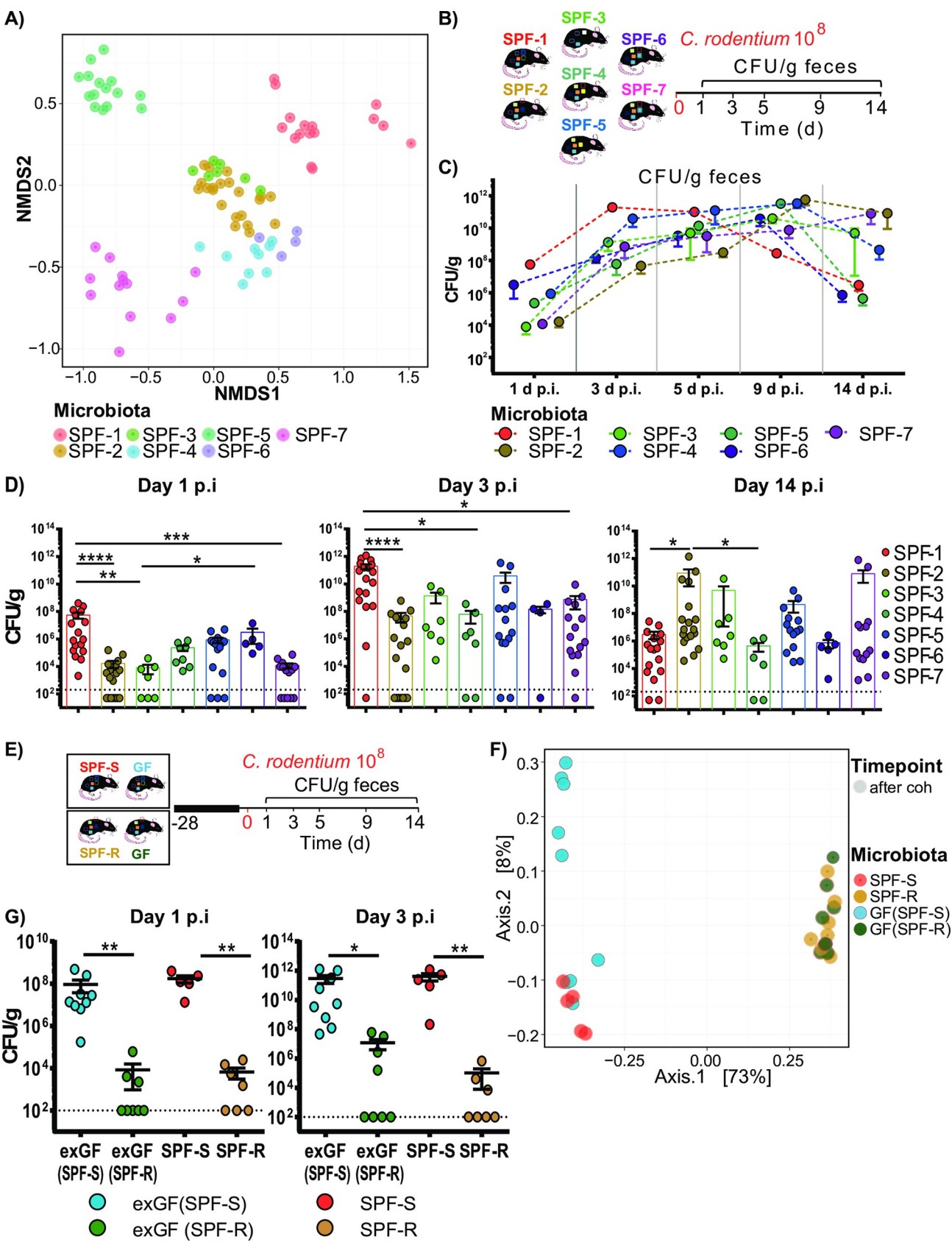

**Fig 1. Isogenic mouse lines feature distinct microbiota compositions and varying disease kinetics after *Citrobacter rodentium* infection.** (A) Fecal bacterial microbiota composition of different specific pathogen free (SPF) mouse lines including SPF-1/SPF-S (n = 17), SPF-2/SPF-R (n = 18), SPF-3 (n = 7), SPF-4 (n = 9), SPF-5 (n = 15), SPF-6 (n = 4) and SPF-7 (n = 12) were evaluated using 16S rRNA gene sequencing. β-diversity was analyzed using Bray-Curtis dissimilarity matrix and non-metric multidimensional scaling (NMDS). (B) Mice with different microbiota settings were orally infected with $10^8$ CFU *C. rodentium*. (C) CFU of *C. rodentium* was determined in the feces of each mouse line at day 1, 3, 5, 9 and 14 post infection (p.i.). (D) Fecal colonization of each individual mouse is indicated after day 1, 3 and 14 p.i. Dashed line indicates detection limit. Results represent one representative experiment with n = 4–17 mice/group as mean ± SEM. P values indicated represent a nonparametric Kruskal-Wallis test with multiple comparisons (one-way ANOVA). *p<0.05, **p<0.01, ***p<0.001, ****p<0.0001. (E) C57BL6/N germ-free mice (GF) were cohoused with SPF-S and SPF-R mice for 4 weeks and then infected with *C. rodentium*. (F) Fecal microbiota was analyzed using 16S rRNA gene sequencing before and after the end of cohousing. β-diversity was analyzed using a principal coordinates analysis (PCoA) plot. (G) *C. rodentium* CFUs in feces of individual mice of each mouse line are displayed on day 1 and 3 p.i. Dashed lines indicate the limit of detection. Results represent one out of three representative experiments with n = 5–9 mice/group as mean ± SEM. *P* values indicated represent a Mann-Whitney U test comparison between group exGF(SPF-S) and exGF (SPF-R) as well as SPF-S and SPF-R. *p<0.05, **p<0.01, ***p<0.001, ****p<0.0001.

lines (Fig 3A). At this time-point, no differences in *C. rodentium* CFU were detected in the stool of both mouse lines suggesting that *C. rodentium* was not immediately killed during early GI passage. To test if observed differences in the fecal colonization are also visible in the intestinal organs, we infected both mouse lines with a bioluminescent *C. rodentium* strain and imaged the infection using an *in vivo* imaging system (IVIS) (Fig 3B). Again, SPF-S mice showed a higher colonization at the early time points (day 1 to 5) of infection, whereas the SPF-R mice were more highly colonized at later time points (day 8 to 12) of infection (Fig 3B). To test, whether the differences at the early time point of infection occur in the luminal content or in the intestinal organs, mice were sacrificed either 1 day or 3 days p.i.. Correlating with the other results, SPF-S mice displayed a significant higher pathogen burden in the intestinal content of the cecum and the colon, as well as in the cecal tissue already after one day and in the small intestinal lumen and tissue after 3 days (Fig 3C and 3D). As differences in colonization between SPF-S and SPF-R mice occur in the cecal lumen, we next assessed colonization *in vitro* to exclude influences of immune cells infiltrating after the infection. For this purpose, we incubated $10^6$ CFU of *C. rodentium* under anaerobic and aerobic conditions in cecal content of SPF-S and SPF-R mice for 6 and 24 hours (Fig 3E). Strikingly, the cecal content of SPF-S mice enabled a significant higher pathogen burden compared to the cecal content of SPF-R mice in both conditions only after 24 but not six hours (Fig 3F). This demonstrates that

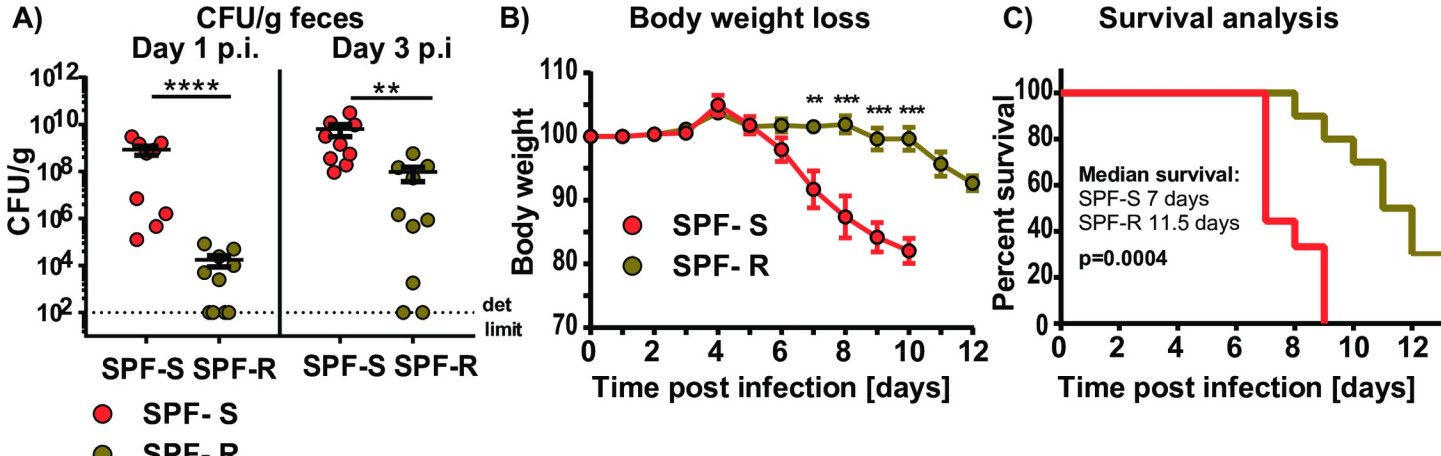

**Fig 2. Variations in microbiota composition are correlated with lethal outcome of infections with a Shiga toxin-expressing *C. rodentium* strains.** (A-C) SPF-S and SPF-R mice were infected orally with $10^8$ CFU *C. rodentium* DBS 770 and fecal colonization, body weight loss and survival were determined. Dashed lines indicate the limit of detection. Results represent two independent experiments with n = 4–5 mice/group as mean ± SEM. P values indicated represent a nonparametric Kruskal-Wallis test *p<0.05, **p<0.01, ***p<0.001, ****p<0.0001.

competition with the microbiota or extended presence of their products in the cecum is sufficient to inhibit the growth of *C. rodentium*.

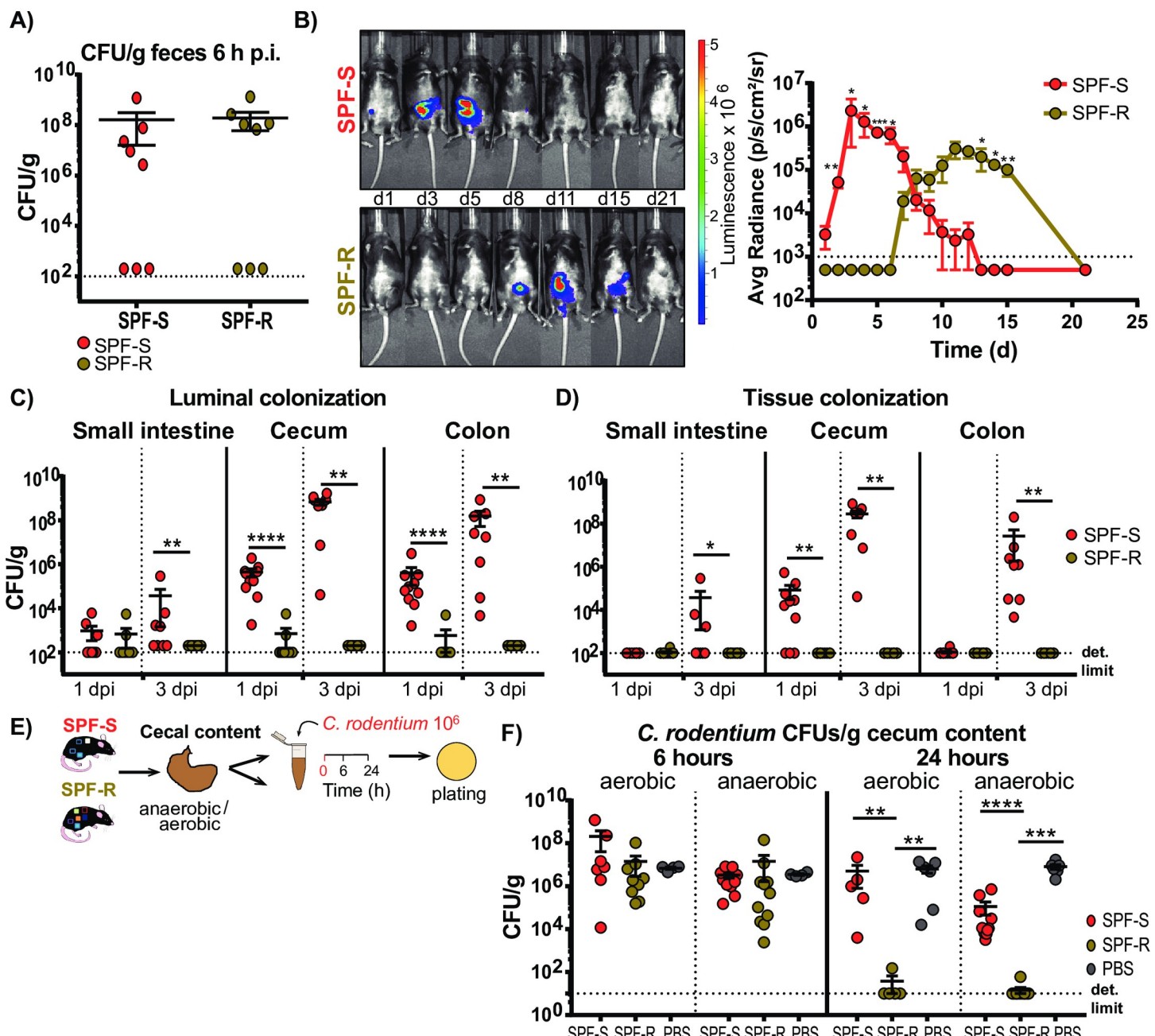

**Fig 3. Delayed colonization in SPF-R mice is visible in all intestinal organs and tissues but most prominent in the cecum.** (A) SPF-S and SPF-R mice were infected orally with $10^8$ CFU *C. rodentium* and fecal colonization was determined after 6 hours post infection (p.i.). (B) SPF-S and SPF-R mice were infected orally with $10^8$ CFU *C. rodentium* and infection was imaged *in vivo* using an IVIS imaging system. Average radiance (p/s/cm$^3$/sr) was determined at day 1, 3, 5, 8, 11, 15 and 21 p.i. (C-D) SPF-S and SPF-R mice were infected orally with $10^8$ CFU *C. rodentium* and sacrificed after 1 day or 3 days p.i. CFU/g organ content and tissue was determined. Dashed lines indicate the limit of detection. Results represent two independent experiment with n = 5–7 mice/group as mean ± SEM. P values indicated represent a nonparametric Kruskal-Wallis test *p<0.05, **p<0.01, ***p<0.001, ****p<0.0001. (E) Isolated cecal content of SPF-S and SPF-R mice was diluted 1:1 in PBS, inoculated with $10^6$ CFU *C. rodentium* and cultivated under aerobic and anaerobic conditions for 24 h before plating on selective agar plates. (E-F) Recovery of *C. rodentium* in isolated cecal content after 6 and 24 h of aerobic and anaerobic cultivation. Dashed lines indicate the limit of detection. Results represent one or two independent experiments with n = 5–7 mice/group as mean ± SEM. *P* values indicated represent a Mann-Whitney U test comparison between groups with **p<0.01, ***p<0.001, ****p<0.0001.

## Cohousing and transfer of cultivable bacteria from resistant SPF-R mice leads to variable protective outcome in SPF-S mice

To identify if bacteria linked to susceptibility or resistance can be transferred and alter susceptibility *in vivo*, SPF-S (susceptible) and SPF-R (resistant) mice were either cohoused for 3 weeks to potentially transfer all bacteria, or received the cultivable fractions of aerobic and anaerobic bacteria isolated from the cecal content of SPF-R mice by oral gavage (Fig 4A and S2 Fig). Subsequently, mice were infected with *C. rodentium* after 3 weeks of microbiota transfer and sacrificed at 3 days p.i. to determine CFUs in the intestinal organs and content. Strikingly, the protective phenotype did not show full penetrance and in all groups. Some recipient SPF-S mice showed a fully protected phenotype, whereas other showed no or only minor reduction in the intestinal CFUs (Fig 4B–4E). Of note, the cohousing and transfer experiments were repeated several times (n = 3 with 4–6 mice per group) with similar outcomes. In each experiment, some mice were fully protected, whereas others had similar CFUs as the untreated SPF-S mice (Fig 4D and 4E). Based on these observations, we hypothesized that the causative bacteria may only be transferred to some mice resulting in the variable protective effect.

## Resistance of SPF-R mice is associated with higher diversity and abundance of bacteria in the Firmicutes phylum

To identify bacteria associated with the protective phenotype after microbiota transfer, all recipient animals were assigned to the classes "susceptible" or "resistant" according to the CFU/g cecal content after 3 days p.i. in the cecum (Fig 5A). All mice below the threshold value of $10^6$ CFU/g cecal content were considered to be "resistant" whereas all animals above the threshold were considered to be "susceptible". Analysis of the beta-diversity from SPF-1 mice receiving anaerobic and aerobic bacteria from SPF-R mice revealed a clustering according to susceptibility (Fig 5B). Moreover, analysis of α-diversity showed significantly elevated species richness in terms of total observed species and in terms of numbers in relation to evenness in the resistant SPF-S mice (Fig 5C) indicating that more bacteria were transferred from the SPF-R microbiota, which harbor a more complex microbiota than the initial SPF-S microbiota (S1 Fig and S3A–S3C Fig). The comparison of relative abundances on the genus level revealed striking differences according to the "susceptible" and "resistant" phenotypes regarding the abundance and diversity of bacteria within the Firmicutes phylum of both groups (anaerobic and aerobic bacteria transfer) (Fig 5D and 5E). A frequent trait of bacteria originating from this phylum is their involvement in the generation of SCFAs such as acetate, propionate and specifically butyrate. We hypothesized that resistance might be associated with a higher ratio of SCFA producing bacteria in the microbiota of SPF-R mice. To support this hypothesis we compared the abundance of bacteria annotated to be involved in the production of butyrate [24]. Indeed, we found significantly elevated levels of bacteria of the genera *Lachnospiraceae*, *Ruminiclostridium*, *Butyricicoccus*, *Intestinimonas*, *Lachnoclostridium*, *Roseburia* and *Ruminicoccaceae* (Figs 5F,5G, 2E and 2F) in the cohoused mice or SPF-S mice receiving anaerobic or aerobic bacteria from SPF-R mice, whereas levels of *Coprococcus* were only elevated in mice receiving anaerobic cultivated bacteria (Fig 5F). The presence of distinct, putatively spore-forming, anaerobes in protected SPF-S mice receiving aerobically cultivated bacteria suggests that viable spores were transmitted. Furthermore, susceptible mice remained a high abundance of *Akkermansia* (Phylum Verrucomicrobia) in their microbiome (Figs 5D and 5E and S2E, S3G and S3H Figs), but *Akkermansia* abundance did not correlate with the phenotype change in SPF-S mice from susceptible to resistant (S3H Fig). Since the transfer of aerobic cultivated bacteria did also result in reduced susceptibility, we hypothesized that altered oxygen availability and a higher abundance and diversity of low abundance taxa such as endogenous

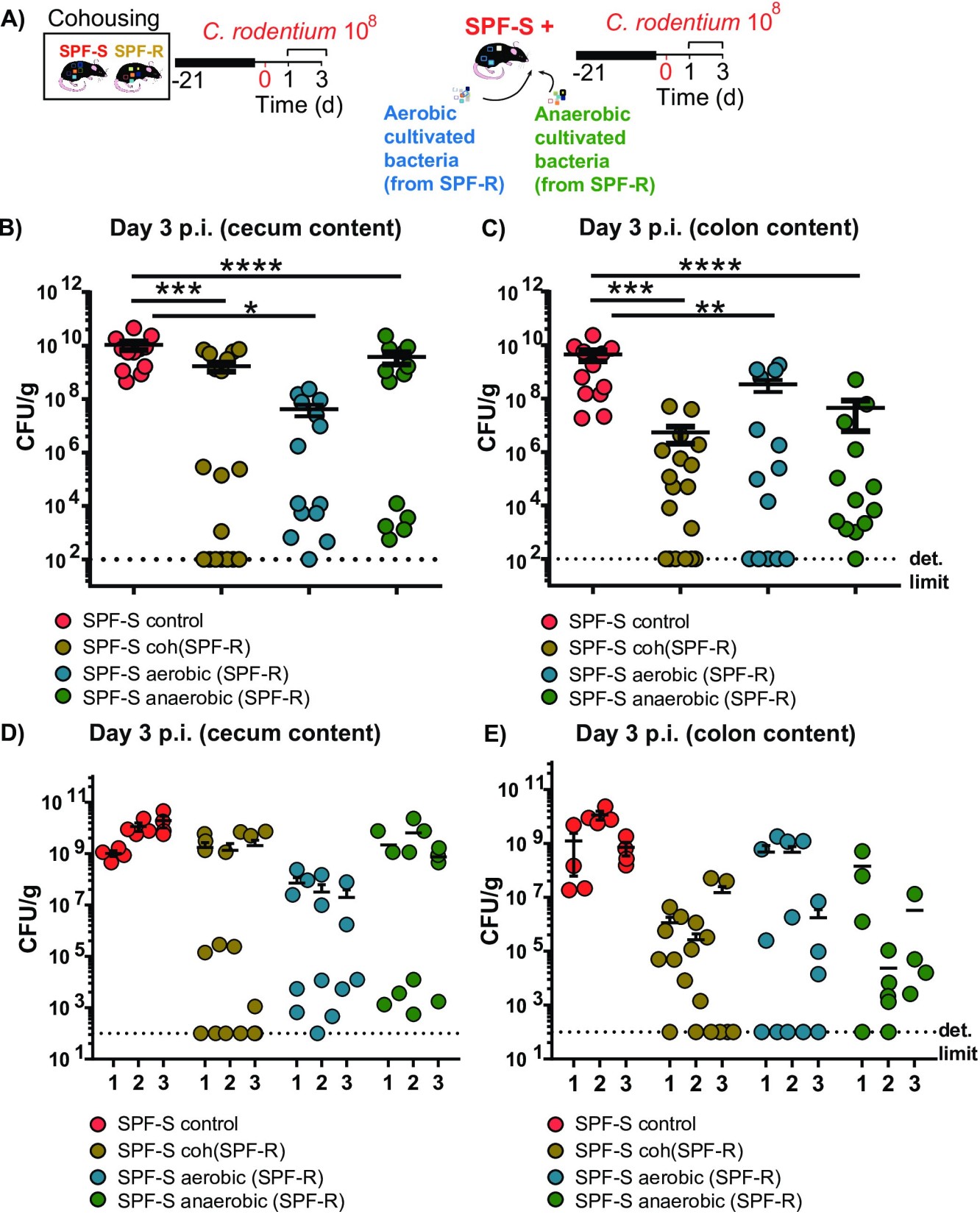

**Fig 4. Cohousing and transfer of cultivable bacteria from resistant SPF-R mice leads to variable protective outcome in SPF-S mice.** (A) Groups of susceptible SPF-S mice were pretreated with different bacteria. One group was cohoused with SPF-R mice for 3 weeks before infection. One group received a mixture of all cultivable aerobic bacteria isolated from cecal content of SPF-R mice. One group received all cultivable anaerobic bacteria isolated from cecal content of SPF-R mice. The last group was left untreated. All mice were infected with $10^8$ CFU *C. rodentium* after 3 weeks and CFUs/g feces and organ content and tissue were assessed after day 1 and 3 p.i. (B-C) Pooled CFUs of *C. rodentium* in the cecal and colon content at day 3 p.i. are displayed. (D-E) CFUs of *C. rodentium* in the cecal and colon content at day 3 p.i. are for each single experiment are displayed. Results represent three independent experiments with n = 4–6 mice per group as mean with SEM. *P* values indicated represent a Mann-Whitney U test comparison between groups with *p<0.05, **p<0.01, ***p<0.001, ****p<0.0001.

*Enterobacteriaceae* and other facultative anaerobic bacteria may also impact the disease susceptibility in this context. Therefore, we first indirectly assessed oxygen availability in the intestinal tissue by measuring the protein expression of HIF1α, a marker for hypoxia in intestinal tissues [25]. Western blot analysis demonstrated that HIF1α was equally expressed in both mouse lines in the cecal tissue at steady state and 3 days p.i. (S4A and S4B Fig) suggesting that hypoxic conditions and oxygen availability in the gut tissue were comparable between both mouse lines. In an alternative approach, fecal and cecal material of both mouse lines was cultured on Mac-Conkey agar plates as endogenous facultative anaerobes are in the gut frequently below the detection limit of sequencing-based approaches (S4C–S4E Fig). Indeed, we could recover significantly more facultative anaerobe bacteria from the cecum and feces of SPF-R mice (S4C Fig). Of note, no Enterobacteriaceae were identified in SPF-S mice, while multiple species including *Citrobacter amalonaticus*, *Escherichia coli* and *Klebsiella oxytoca* were isolated from SPF-R mice (S4D and S4E Fig). Nevertheless, transfer of these bacteria into SPF-S mice did not result in significantly reduced *C. rodentium* CFUs in the cecum and colon 3 days p.i. (S4G and S4H Fig), suggesting that the availability of oxygen close to the mucosa and the variable abundance of facultative anaerobes are not responsible for the different susceptibility of SPF-S and SPF-R mice respectively.

## The metabolomics profile of resistant SPF-R is characterized by elevated SCFA levels, which strongly impair the growth of *C. rodentium in vitro* at pH 6.0

Based on the microbial signatures in the resistant mice linking SCFA producing bacteria to the resistant phenotype, we next analyzed metabolome profiles in both mouse lines, particularly the levels of SCFAs. To characterize differences in the cecal metabolome in SPF-S and SPF-R mice at steady state and after infection, we performed targeted (for SCFA) and non-targeted metabolomics (Fig 6A and S5 Fig). Strikingly, SPF-R mice showed significantly elevated levels of butyrate in the cecum with an average concentration of 80 nmol/mg cecal content in contrast to levels around 20 nmol/mg cecal content in SPF-S mice at steady state and after infection. In addition, acetate was elevated in SPF-R mice at both time points, but the fold-difference was smaller as for butyrate. Propionate levels were also slightly increased in resistant SPF-R mice before and after infection (Fig 6A). Using non-targeted metabolomics, 70 out of 247 compounds could be successfully identified. A non-supervised cluster analysis revealed a strong separation of both mouse lines based on the intestinal metabolome profiles. The levels of 26 metabolites were already significantly different at steady state prior infection (S5A and S5C Fig). From these, 20 metabolite levels were significantly elevated in SPF-1 mice compared to SPF-2 mice, including different citric acid cycle intermediates or fermentation products such as succinate, fumarate and malate, urea-cycle and poly amine metabolism intermediates such as urea, putrescine and ornithine as well as the amino alcohol ethanolamine and the flavone apigenin (S5A and S5C Fig). In contrast, only six metabolites, mostly amino acids were elevated in resistant SPF-R mice. Several metabolites enriched in susceptible SPF-S mice (i.e.

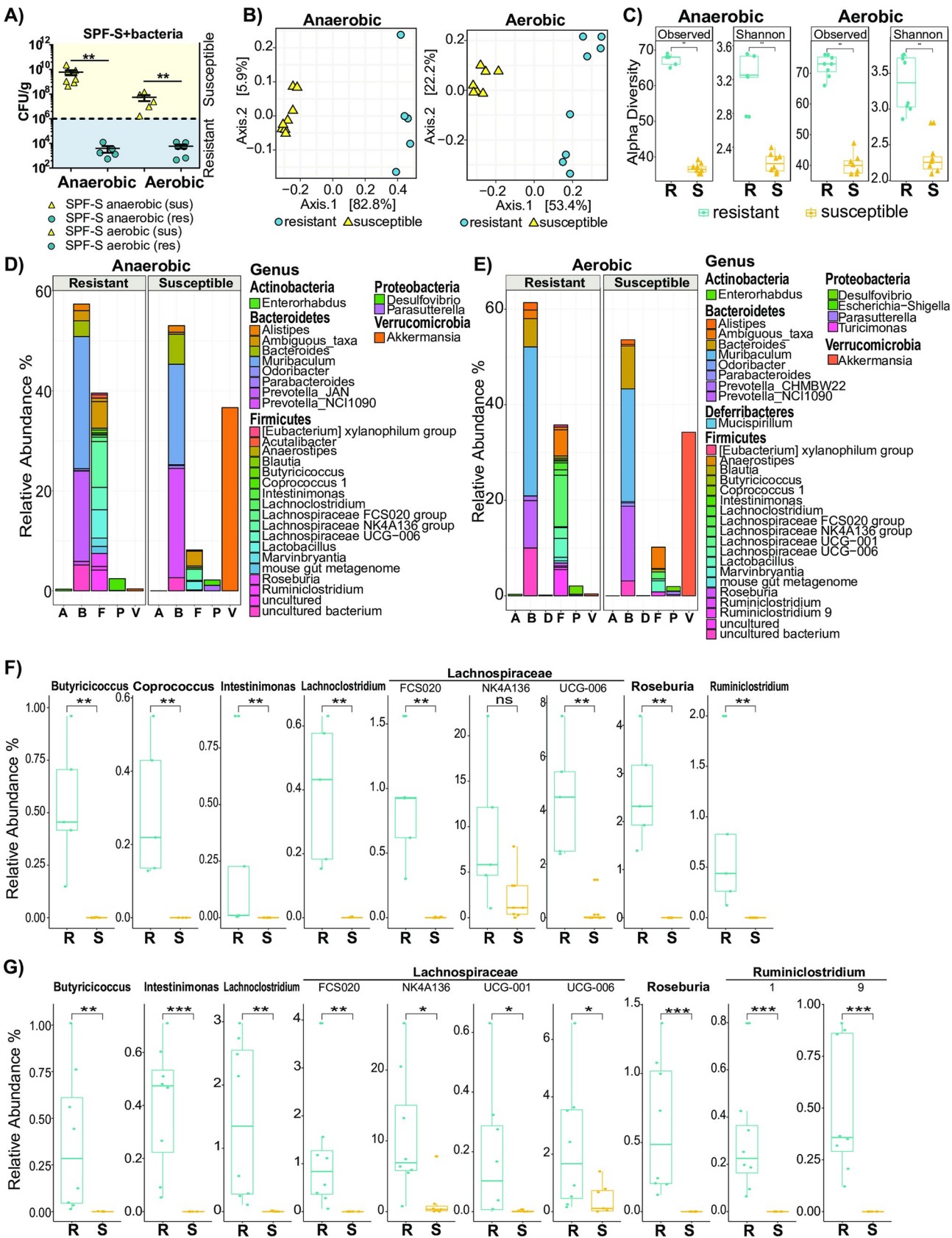

**Fig 5. Resistance of SPF-S mice is associated with higher diversity and abundance of bacteria in the Firmicutes phylum.** (A) Assignment of SPF-S mice receiving a cultivated fraction of anaerobically or aerobically cultivated bacteria from the SPF-R mice to the class "resistant" and "susceptible" according to the cecal CFUs of *C. rodentium* after day 3 p.i.. A threshold of $10^6$ was used for discrimination of both groups. (B) Fecal microbiota of resistant and susceptible SPF-S mice of the "anaerobically treated" and "aerobically treated" group was analyzed using 16S rRNA gene sequencing after cohousing using a principal coordinates analysis (PCoA) plot (C) α-diversity was determined using Chao1 and Shannon index. P values indicated represent a non-parametric Wilcoxon signed rank test *p<0.05, **p<0.01, ***p<0.001, ****p<0.0001. (D-E) Fecal microbiota of resistant and susceptible "anaerobically treated" and "aerobically treated" SPF-S mice were analyzed using 16S rRNA gene sequencing. Relative abundances of bacterial genera are shown and grouped according to their phylum. Bars represent the mean of all mice within the group. Representative data derived from three independent experiments are shown. (F-G) Relative abundance of significantly different SCFA producing members of treated SPF-S mice of the genera *Butyricicoccus*, *Coprococccus*, *Intestinimonas*, *Lachnoclostridium*, *Lachnospiraceae*, *Roseburia* and *Ruminoclostridium* are shown. P values indicated represent a non-parametric Wilcoxon signed rank test *p<0.05, **p<0.01, ***p<0.001.

succinate, fumarate and malate) were sufficient to enable *in vitro* growth of *C. rodentium* as sole carbon sources in minimal media suggesting they could support *C. rodentium* growth early on *in vivo*. Of note, at one day after infection we observed only minor changes in microbiome metabolite levels in both mouse lines indicating that the early time point of infection did not change the metabolic landscape dramatically in both mouse lines compared to the steady state conditions (S5B and S5D Fig).

Based on the strongly altered SCFA concentrations and their previous association to growth inhibition of *Enterobacteriaceae* such as *Salmonella* Typhimurium [5], *E. coli* and *K. pneumoniae* [26], we tested the inhibitory properties of different SCFAs against *C. rodentium in vitro*. To do so, we added different concentrations of butyrate and acetate ranging from 10 mM to 100 mM and for propionate ranging from 1 mM to 8 mM to BHI medium. The pH values were normalized to a slightly acidic pH of 6.0 and a neutral pH of 7.0. Concentrations of SCFAs and pH approximately represent biological conditions measured in the mouse lines at steady state *in vivo* (Fig 6A and S2F Fig). Measuring the OD of *C. rodentium* over time, we noticed significantly impaired growth at higher concentrations of butyrate at a pH value of 6.0, whereas acetate and propionate did only partially reduce the growth of *C. rodentium* at pH 6.0 at high physiological concentrations. No significant effect for any SCFA was visible at neutral pH in line with findings on other *Enterobacteriaceae* [26] (Fig 6B). These results indicated that higher concentrations of SCFAs, especially butyrate, as seen in the SPF-R mice are able to abrogate growth of *C. rodentium in vitro*. Since SCFAs are present as a mixture under *in vivo* conditions, we tested different concentrations and combinations of the SCFAs as well approximately matching mixtures of all three SCFAs found in SPF-S and SPF-R mice against *C. rodentium in vitro*. First, we defined butyrate to be the main factor responsible for inhibition of *C. rodentium* growth when administered in concentrations higher than 50 mM (Fig 6C). Moreover, butyrate was even more effective when administered with 4 mM propionate or concentrations of acetate (30 mM and higher), whereas acetate and propionate alone were not sufficient to inhibit growth of *C. rodentium* completely even at the highest tested concentration (Fig 6B and S6 Fig). Finally, all three SCFAs tested together, the SPF-R mice related mixture of SCFAs lead to nearly complete abrogation of *C. rodentium* growth, whereas the SPF-1 related mixture of SCFAs did only have a minor impact on growth (Fig 6D).

## Supplementation of SPF-S mice with SCFA is sufficient to reduce *C. rodentium* growth *in vitro* as well as *in vivo*

Next, we wanted to test, whether an increase of the SCFA concentration in the SPF-S cecal content similar to the levels found in the SPF-R cecal content would be sufficient to achieve the protected phenotype *in vitro* and *in vivo*. First, isolated cecal content of SPF-S mice was supplemented with 80 mM of butyrate and adjusted to a pH of 6.0 and inoculated with $10^6$ CFUs of *C. rodentium* and incubated for 24h at 37°C. As a comparison, untreated SPF-1 and

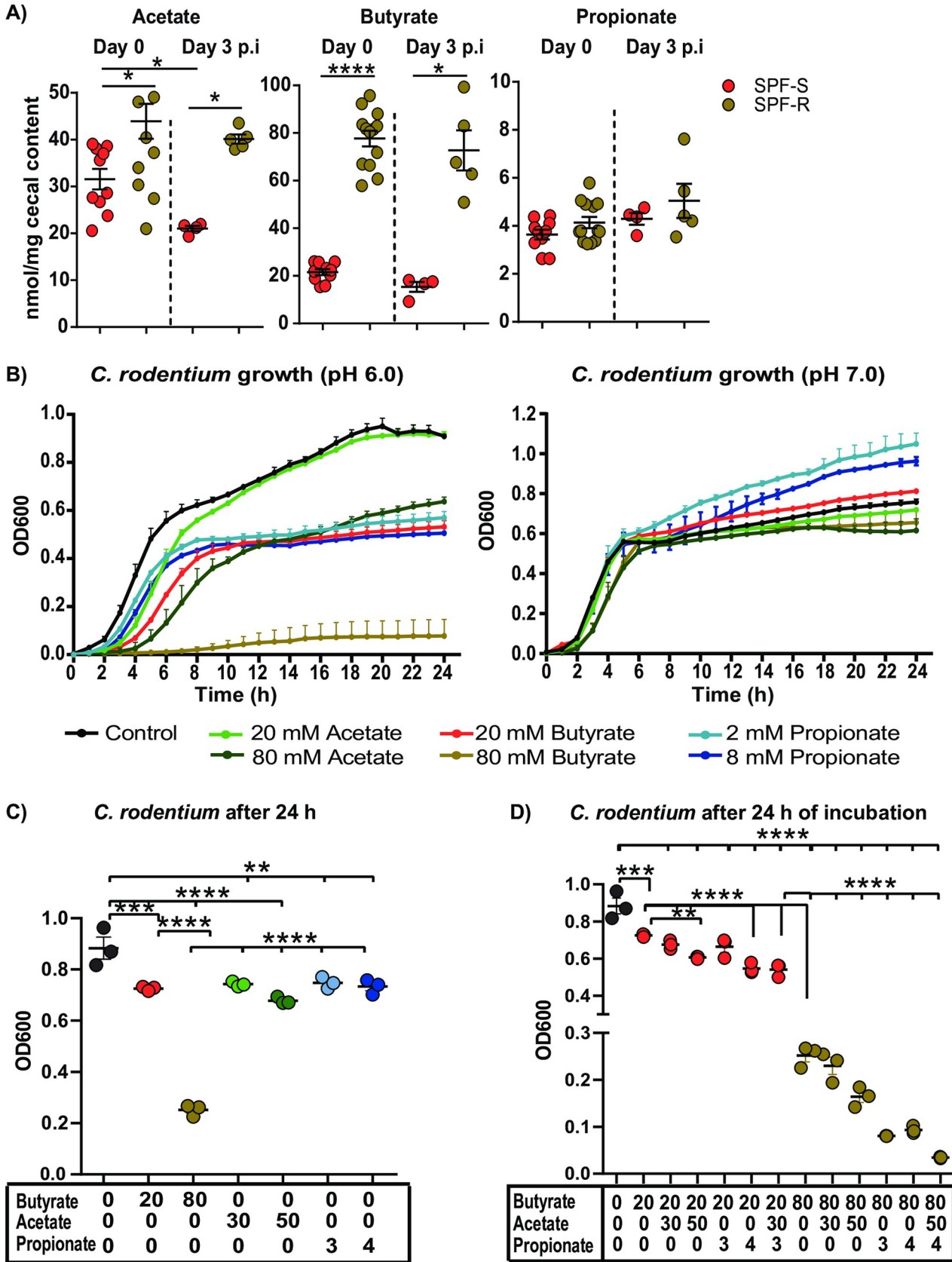

**Fig 6. The metabolome of resistant SPF-R is characterized by elevated SCFA levels, which strongly impair the growth of *C. rodentium in vitro* at pH 6.0.** (A) Cecal SCFA levels of butyrate, acetate and propionate for SPF-S and SPF-R mice at steady state and day 3 p.i. are displayed in nmol/mg cecal content. Pooled results of two independent experiments as mean ± SEM are shown. *P* values indicated represent a Mann-Whitney U test comparison between groups with *p<0.05, ****p<0.0001. (B) *C. rodentium* growth displayed as optical density (OD) over time at pH 6.0 (left) or pH (7.0) in BHI medium supplemented with different concentrations of acetate, butyrate or propionate or without any SCFA added. (C-D) *Citrobacter* growth displayed as optical density (OD) over time at pH 6.0 in BHI medium supplemented with different mixtures of acetate, butyrate and propionate or without any SCFA added. (C-D) One data point represents mean value of three replicates. Values out of three independent experiments are displayed. *P* values indicated represent a One-way ANOVA between groups with **p<0.01, ***p<0.001, ****p<0.0001.

SPF-R cecal content was spiked with the same number of bacteria. Recovery of viable *C. rodentium* colonies after 24 hours revealed that the butyrate supplementation of the cecal content was sufficient to achieve an inhibitory effect comparable to the one observed in SPF-R cecal content, indicating that the total amount of butyrate is as effective as the presence of specific bacteria for the protection (Fig 7A and 7B). Next, to verify these findings *in vivo*, we supplemented susceptible SPF-S mice with a SCFA mixture consisting of 150 mM butyrate, 150 mM acetate and 30 mM propionate in the drinking water (Fig 7C). Mice received SCFAs one day before infection until 3 days p.i.. Comparison of the CFUs in the cecum revealed significantly decreased CFUs in the SCFA supplemented SPF-S mice compared to untreated SPF-S mice (Fig 7D). Yet, in contrast to the SPF-R mice, mean CFUs were still significantly elevated. This coincided with only a partial restoration of protective SCFA concentration in the cecum after oral SCFA supplementation of SPF-S mice, indicating that sufficiently high concentrations of SCFAs have to be reached *in vivo* for the protection (Fig 7E). Of note, only 5 out of 19 animals reached the threshold of 50 nmol/mg butyrate which was required to abrogate *C. rodentium* growth *in vitro*. Those five responding animals achieved a fully protected phenotype similar to the SPF-R mice (Fig 7G). To exclude that those 5 animals had a different microbiota composition compared to the other animals tested, we assessed the microbiome composition before SCFA supplementation and at the end of the experiment at 3 days p.i. (S7 Fig). The responding animals did not show pronounced microbiota differences before the start of SCFA supplementation in terms of species composition (S7A and S7C Fig) or species richness and evenness (S7B Fig). After 3 days p.i. similar microbiome compositions were observed in all three groups of SPF-S mice, except the abundance of *Enterobacteriaceae* (S7G Fig). Responder animals show significantly less *C. rodentium* compared to non-responding animals and control animals (S7H Fig). Microbiome data supported the hypothesis that variations in the butyrate concentrations lead to changes in *C. rodentium* abundances rather than variability in the microbiota composition of the susceptible SPF flora. Furthermore, SCFA supplementation was sufficient to reduce the pH value to similar levels as found in resistant SPF-R mice (Fig 7F and 7I). To assess the effects of butyrate specifically in the absence of the present bacteria in the SPF-S flora we performed the *in vivo* butyrate supplementation in germ-free and antibiotic-treated SPF-S mice (Fig 7J and 7L). In both setups, mice received butyrate one day before infection until 3 days p.i. In both experiments, CFUs in the cecum were partially reduced in the SCFA supplemented animals (Fig 7K and 7M). Corresponding SCFA values in the cecum were significantly elevated in both setups, but were overall low compared to naïve SPF-S conditions (S7D and S7F Fig). The cecal pH value was comparable between the untreated and treated animals but tended to be reduced in SCFA supplemented mice (S7E and S7G Fig). These experiments support the key role of butyrate to reduce the CFUs of *C. rodentium* under *in vitro* and *in vivo* conditions in absence of other resident microbes.

Taken together, these results validate that higher abundance and diversity of SCFA-producing bacteria in the cecum of SPF-R mice lead to elevated butyrate levels and reduced pH value in the lumen of the cecum, thereby efficiently reducing growth of *C. rodentium* at the early phase of infection.

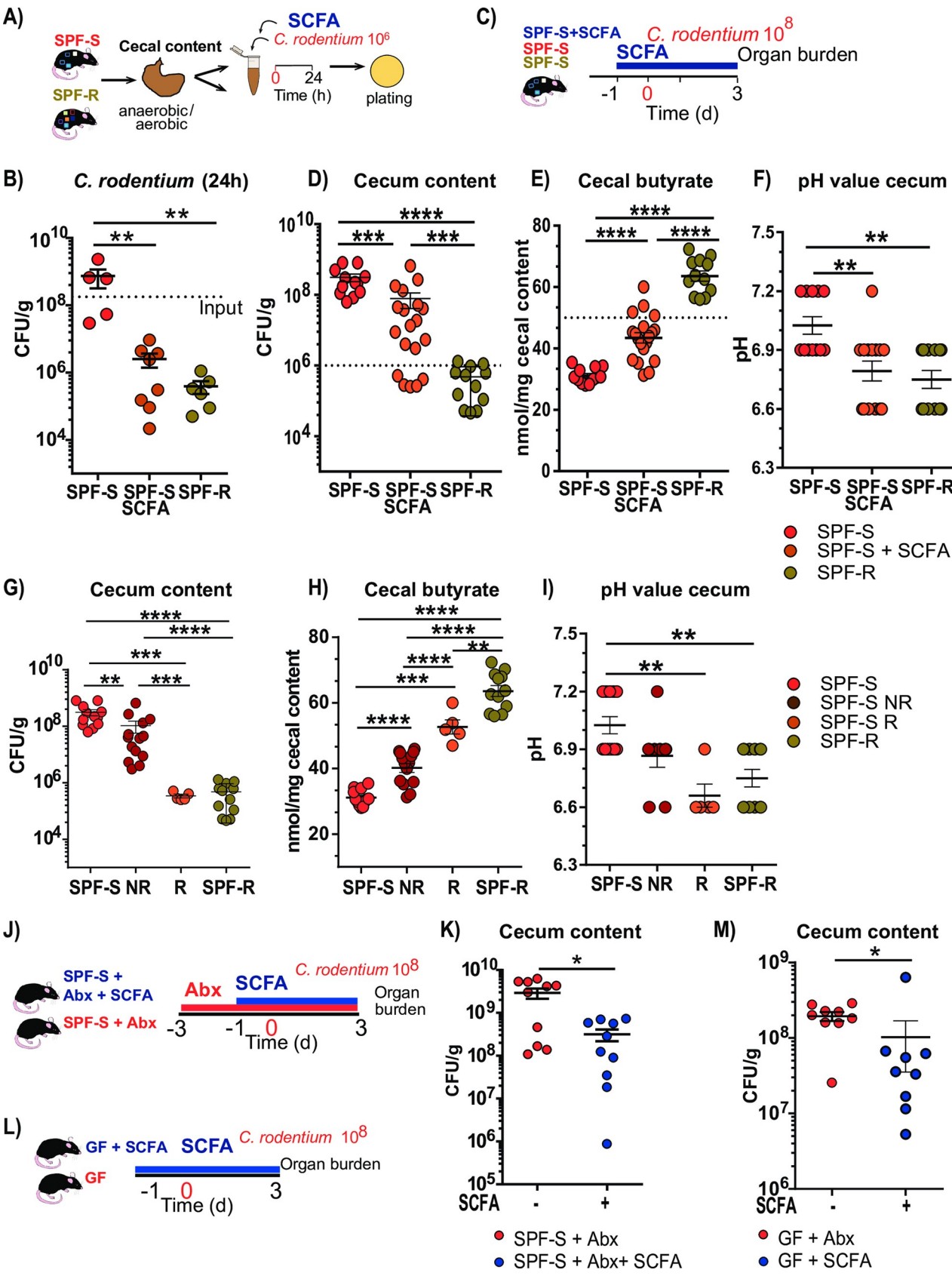

**Fig 7. Supplementation of susceptible SPF-S mice with short-chain fatty acids lead to reduced *C. rodentium* burden *in vitro* and *in vivo*.** (A-B) Recovery of *Citrobacter* after 24 h anaerobic incubation in cecal content of SPF-S mice, SPF-S mice supplemented with 80 mM of butyrate, 50 mM acetate and 4 mM Propionate and SPF-R mice content. *P* values indicated represent a Mann-Whitney U test comparison between groups with **p<0.01. Dashed line indicates input level of *C. rodentium* (C-D) SPF-S mice were supplemented with 150 mM butyrate, 150 mM acetate and 30 mM propionate in the drinking water one day before infection until three days post infection. Untreated SPF-S and SPF-R as well as SCFA supplemented SPF-S mice were infected orally with $10^8$ CFU *C. rodentium* and sacrificed three days post infection to assess colonization. (D) *C. rodentium* CFUs in cecal lumen of individual mice of each group are displayed after day 3 p.i. Dashed line represents threshold level for resistant animals. (E) Cecal butyrate level of each group is displayed after day 3 p.i.. Dashed line represents threshold level for resistant animals. (F) Cecal butyrate level of each group is displayed after day 3 p.i. (G-I) Cecal CFUs, cecal butyrate level as well as cecal pH values are displayed for all groups. SCFA supplemented animals are grouped based on the CFU level for *C. rodentium* in the cecum after day 3 p.i.. Results represent three pooled experiments with n = 4–7 mice/group as mean ± SEM. *P* values indicated represent a Mann-Whitney U test comparison between groups with **p<0.01, ***p<0.001, ****p<0.0001. (J) Antibiotic treated SPF-S mice were supplemented with 150 mM butyrate in the drinking water one day before infection until three days post infection. (K) Cecal CFUs are displayed after day 3 p.i. (L) C57BL6/N germ-free mice were supplemented with 150 mM butyrate in the drinking water one day before infection until three days post infection. (M) Cecal CFUs are displayed after day 3 p.i. Results represent two pooled experiments with n = 5–9 mice/group as mean ± SEM. *P* values indicated represent a Mann-Whitney U test comparison between groups with **p<0.01, ***p<0.001, ****p<0.0001.

## Discussion

It is widely acknowledged that the intestinal microbiota has enormous impact on the individual susceptibility against invading pathogens and the severity of intestinal inflammation via direct and immune-mediated mechanisms [1]. Especially reduction of the colonization resistance against certain *Enterobacteriaceae* such as *Salmonella* [5], *E. coli* or *K. pneumoniae* [26] after antibiotic treatment has been linked with a reduction of SCFA-producing bacteria. In the present study, we focus on the impact of naturally occurring variations in the microbiota composition of laboratory mice and the influence of specific members in the microbiota in murine *C. rodentium* infection without any antibiotic interventions. The *C. rodentium* models have been used intensively to model human infections with EHEC and EPEC and to dissect direct and immune-mediated mechanisms of the microbiota influencing susceptibility and severity. For example, it has been shown that antibiotic treatment renders mice more susceptible to *C. rodentium* infections [13]. Also, the presence or absence of bacteria on a general level such as elevated Bacteriodetes over Firmicutes level [14] or the presence of a single kind of bacterium such as segmented-filamentous bacteria (SFB) [15] or higher ratios of Clostridia [16] have been associated with elevated resistance toward *C. rodentium* via immune-mediated mechanisms. In contrast, the overall absence of microbes leads to enhanced susceptibility and persistence of *C. rodentium* in germ-free mice [17]. Also direct mechanisms of commensal bacteria such as expansion of commensal *E. coli* are able to inhibit *C. rodentium* colonization by competition for monosaccharides as a nutrient source [17]. Following up on the observation that mice from different breeding facilities show different susceptibility in related gastrointestinal disease models such as *Salmonella* Typhimurium [19,20] or DSS-induced colitis ]18] due to presence or absence of specific members of the microbiota, we decided to characterize the impact of the microbiota composition during *C. rodentium* infection, since knowledge about specific species and pathogen-commensal interactions that contribute to protection against colonization of *C. rodentium* remains elusive.

Indeed, striking differences regarding the composition of the microbiota and the colonization with *C. rodentium* could be observed in genetically similar C57BL6/N sub-lines already at 1 day p.i.. To exclude any influence of genetic differences or spontaneous mutations reported in C57BL/6N sublines [27,28], we transferred one susceptible (SPF-S) and resistant (SPF-R) microbiota to genetically identical germ-free recipient mice. Fecal microbiota composition as well as disease progression was highly similar between formerly germ-free mice and their respective cohousing partners, further supporting that the phenotype is fully dependent on the microbiota.

Since *C. rodentium* is a mouse-specific pathogen, which is used to mimic the infections of human with EHEC and EPEC, we were interested to see whether delay in colonization would also translate into prolonged survival using a more pathogenic *C. rodentium* strain. Therefore we used a strain that is able to express Stx [12], which is responsible for intestinal damage and life-threatening systemic diseases such as renal failure in EHEC infections [29]. Again, SPF-S mice were less colonized after one and three days post infection and in line with these results showed significantly reduced body weight loss and prolonged survival.

For all infection experiments, differences in colonization were initially and most strongly detected in the cecal lumen at the early phase of infection at could be also recovered in isolated cecal content indicating that specific bacteria in the cecum can delay colonization early on via direct or indirect mechanisms. Comparing non-cohoused and cohoused mice, we identified that distinct features of the intestinal microbiota are associated with lower *C. rodentium* colonization in early phase of infection. Specifically, SCFA producing bacteria [24] within the Firmicutes phylum such as *Intestinimonas*, *Lachnoclostridium*, *Roseburia* and *Ruminiclostridium* were significantly elevated, whereas the mucin-degrading bacterium *Akkermansia* was only present in higher amounts in SPF-S mice. The presence of *Akkermansia* has mostly been associated with beneficial properties and has been inversely correlated with several diseases including IBD, acute appendicitis, obesity and diabetes [30]. Of note, these diseases may effect integrity or thickness of the mucus layer, thereby influencing abundances of *Akkermansia*. In this study, comparison of the mucus layer and integrity could not reveal significant differences between both mouse lines. Also, the abundance of *Akkermansia* after cohousing did not correlate with changes in the phenotype of the mice suggesting that differences of *Akkermansia* abundances may not sufficiently explain differences in susceptibility against *C. rodentium* in this study.

Further analysis of the 16S rRNA gene profiles of resistant SPF-S animals showed a clear clustering apart from the microbiome of susceptible mice characterized by overall increased species richness most pronounced in the abundance and diversity within the Firmicutes phylum indicating that indeed the presence of specific SCFA producing bacteria is responsible for resistance against *C. rodentium* infection. Further comparison of the metabolomics landscape revealed striking differences between SPF-S and SPF-R mice with a strong increase of succinate, malate and fumarate as well as urea cycle associated metabolites ornithine, putrescine and urea in the susceptible mice. Of note, SPF-R mice harbored significantly more and more divers *Enterobacteriaceae* in the cecum, which might consume metabolites that were elevated in the SPF-S environment thereby decreasing possible carbon sources for *C. rodentium* in the resistant SPF-R mice. Recently, a higher ratio of endogenous *Enterobacteriaceae* naturally occurring in the microbiota of laboratory animals have been associated with reduces susceptibility towards *Salmonella* Typhimurium [20]. While *Enterobacteriaceae* were absent in SPF-S mice, multiple species belonging to this family were isolated from SPF-R mice. However, transfer of those bacteria isolated from the SPF-R to SPF-S mice did not significantly reduce CFUs in the cecum, suggesting that those bacteria did not play a major role in this setting. Furthermore, susceptible SPF-S mice harbored significantly higher levels of Bacteroides species in the microbiome, which are known to increase metabolites involved in gluconeogenesis [31]. This modulation of the metabolite environment has been linked with enhanced EHEC virulence gene expression by an increase of A/E lesion formation through the transcription factor Cra, tightly regulated by sugar concentrations [31]. In addition, the compound ethanolamine was significantly elevated in susceptible mice known to serve as a carbon source for many related bacterial pathogens including *Salmonella*, *Escherichia* or *Klebsiella* [32]. In the case of *Salmonella* Typhimurium, decrease in luminal butyrate levels by eradication of butyrate producing bacteria through streptomycin treatment increased epithelial oxygenation and aerobic expansion of the pathogen [33]. Similarly, increased oxygenation has also been shown to

enhance mucosal *Citrobacter* colonization [34]. Furthermore, *Salmonella* can take growth advantage from consumption of non-fermentable carbon sources such as succinate [35] or 1,2-propane-diol [36], whereas growth can be inhibited by microbiota derived SCFAs by acidification of the intracellular cytoplasm [5]. To exclude that oxygen availability plays a key role in the observed phenotype we assessed hypoxia inducible factor (HIF1-α) [25] expression in the gut tissues of both mouse lines at steady state and 3 days p.i., which did not reveal significant differences between both mouse lines.

Our findings suggest that natural occurring variations in the microbiota without any antibiotic intervention directly changes the susceptibility to gastrointestinal pathogens. These preexisting differences in the microbial environment such as presence and abundance of butyrate and succinate producers explain strongly differing disease susceptibility towards *C. rodentium* in isogenic mice which might also be true for occurring variations in humans infected with a standardized EPEC inoculum [7,8]. Further focusing on the mechanism of the suppression, we tested the inhibitory properties of butyrate on *C. rodentium* by supplementing BHI medium with different concentrations of butyrate, acetate and propionate individually and combined in physiological concentrations as in the cecum of the different mouse lines. Strikingly, only butyrate as well as the mixture of SCFAs found in the SPF-R mice was able to nearly completely abrogate the growth of *C. rodentium* at pH 6.0, but not at pH 7.0, indicating that not only the amount of SCFAs found in the cecum of SPF-R mice but also the reduced cecal pH is responsible for inhibition of *C. rodentium*. In contrast to the inhibitory effects of SCFAs on *C. rodentium* shown in this manuscript, SCFAs have been shown to specifically promote the adherence and motility of EHEC *in vitro* by activation of the expression of locus of enterocyte effacement [37], or intimate attachment and type III secretion [38]. Taken together, the role of SCFAs seems to be multifactorial and differential to the bacterial species present in the gut. Finally, we could demonstrate that supplementation of isolated cecal content of SPF-S mice *in vitro* as well as butyrate supplementation of SPF-S mice *in vivo* effectively reduced the colonization of *C. rodentium* accompanied by a reduction of the cecal pH value to the same extend as found in the resistant SPF-R mice when reaching a critical value of more than 50 nmol/mg in the cecum. Furthermore, butyrate supplementation in antibiotic depleted and germ-free animals also led to significantly reduced cecal CFUs highlighting the impact of SCFAs in this setting. Nevertheless, overall butyrate levels could not be restored by SCFA supplementation alone highlighting the importance of resident microbes to achieve fully protective SCFA levels. Based on these findings SCFA concentrations may serve as a biomarker to predict disease outcome after infection with certain *Enterobacteriaceae*.

In summary, our study demonstrated that presence, abundance and diversity of SCFA producing bacteria strongly influenced disease kinetics and severity of *Citrobacter*-mediated disease. Furthermore, we could identify butyrate as a key metabolite for protection against *C. rodentium*. As previous studies have largely focused on immune-mediated mechanisms, however, our experiments demonstrated that natural variations in the microbiota strongly shape the metabolite environment and intestinal pH value, thereby contributing to naturally occurring variations in the susceptibility against enteropathogens such as *C. rodentium*. These findings highlight the need for microbiota normalization across laboratory mouse lines for infection experiments with the model-pathogen *C. rodentium* independent of investigations of diet and antibiotic usage.

## Methods and subject details

### Ethics statement

All animal experiments have been performed in agreement with the guidelines of the Helmholtz-Zentrum für Infektionsforschung, Braunschweig, Germany, the national animal

protection law (Tierschutzgesetz (TierSchG)) and animal experiment regulations (Tierschutz-Versuchstierverordnung (TierSchVersV)), and the recommendations of the Federation of European Laboratory Animal Science Association (FELASA). The study was approved by the Lower Saxony State Office for Nature, Environment and Consumer Protection (LAVES), Oldenburg, Lower Saxony, Germany; permit No. 33.4-42502-04-14/1415, No. 33.19-42502-04-16/2124 and No. 33.19-42502-04-17/2573.

## Mice

C57BL/6N SPF-1 mice were purchased from NCI and maintained (including breeding and housing) at the animal facilities of the Helmholtz Centre for Infection Research (HZI) under enhanced specific pathogen-free (SPF) conditions. C57BL/6N SPF-2, SPF-3, SPF-4, SPF-6 and SPF-7 mice were purchased from different vendors and housed under enhanced SPF conditions at the HZI for at least two weeks before the start of the experiment: Charles River (SPF-2 and SPF-3), Harlan (SPF-4), Janvier (SPF-6), and Taconic (SPF-7). SPF-5 mice were generated, bred and housed under enhanced SPF conditions at the HZI [39]. Germ-free C57BL/6NTac mice were bred in isolators (Getinge) in the germ-free facility at the HZI. Animals used in experiments were gender and age matched. Female and male mice with an age of 8–12 weeks were used. Sterilized food and water ad libitum was provided. Mice were kept under strict 12-hour light cycle (lights on at 7:00 am and off at 7:00 pm) and housed in groups of up to six mice per cage. All mice were euthanized by asphyxiation with $CO_2$ and cervical dislocation.

## Bacterial strains

Nalidixic acid and kanamycin-resistant and bioluminescent *Citrobacter rodentium (C. rodentium)* strain ICC180 [23] was used for most infection experiments and *in vitro* assays. Genetically modified Shiga toxin-producing and chloramphenicol-resistant *C. rodentium* strain DBS770 [12] was used for infection experiments under BSL3 conditions.

## Microbiota manipulation

For cohousing experiments with conventional raised mice, age- and gender-matched C57BL/6N mice were housed together in cages at 1:1 ratios for three to four weeks before infection experiments. For cohousing experiments with germ-free mice, mice were cohoused at 1:1 ratio in specific ventilated isocages equipped with individual HEPA filters (Techniplast).

## *In vivo C. rodentium* infection

Bioluminescence expressing *C. rodentium* strain ICC180 was used for all infection experiments [23]. Shiga toxin-producing *C. rodentium* strain DBS770 [12] was used for two infection experiment under BSL3 conditions. *C. rodentium* inoculi were prepared by culturing bacteria overnight at 37°C in LB broth with 50 µg/ml kanamycin. Subsequently, the culture was diluted 1:100 in fresh medium, and subcultured for 4 hours at 37°C in LB broth [19]. Bacteria were washed twice in phosphate-buffered saline (PBS). Mice were orally inoculated with $10^8$ CFU of *C. rodentium* diluted in 200 µl PBS. Weight of the mice was monitored, and feces were collected at different time points (1, 3, 5, 9 and 14 days) after infection for measuring pathogen burden and 16S rRNA sequencing. Mice were infected with bioluminescence expressing *C. rodentium* as described. After different timepoints of the infection (d1, 3, 5, 8, 11, 15, and 21) mice were anesthetized using isoflurane (4 vol %) and transferred into the in vitro imaging system (IVIS). Before the first imaging, mice were shaved on the belly to enhance detection of bioluminescence. Images were taken after different exposure times (1 s, 30 s, 1 min, 3 min)

and luminescence was quantified as average radiance (p/s/cm3/sr). After termination of the experiment luminescence intensities of all pictures were normalized and quantified using the Live Image software.

## Quantification of fecal *C. rodentium* colonization

Fresh fecal samples were collected, and weight was recorded. Subsequently, fecal samples were diluted in 1 ml LB medium and homogenized by bead-beating with 1 mm zirconia/silica beads for two times 25 seconds using a Mini-Beadbeater-96 (BioSpec). To determine CFUs, serial dilutions of homogenized samples were plated on LB and MacConkey plates with 50 μg/ml kanamycin. Plates were cultured at 37˚C for 1 day before counting. CFUs of *C. rodentium* were calculated after normalization to the weight of feces.

### *In vitro* assays

Mice were sacrificed and cecal content was isolated, weighted and diluted in a 1:1 ratio with PBS or BHI and homogenized for two times 25 seconds using a Mini-BeadBeater-96 (BioSpec). Samples were either prepared aerobically or anaerobically in an anaerobic chamber. If required SCFA were added in different concentrations and pH was adjusted to pH 6.0 or 7.0 using HCl or NaOH. Bacteria were grown in appropriate media and normalized to $10^6$ CFUs. Tubes were inoculated with 10 μl *C. rodentium* and cultivated at 37˚C under aerobic or anaerobic conditions for 6 or 24 hours. Twenty-five μl of each sample were serial-diluted in 96 well plates and plated on selective agar plates with kanamycin to recover viable amounts of *C. rodentium*.

## Growth curve assays

*C. rodentium* culture was grown overnight at 37˚C in LB broth with 50 μg/ml kanamycin. Subsequently, the culture was diluted 1:100 in fresh medium, and subcultured for 4 hours at 37˚C in BHI broth. A 96-well plate was equipped with 190 μl of BHI medium normalized to pH 6.0 or 7.0 with or without short-chain fatty acids added. *C. rodentium* culture was normalized to $10^6$ CFUs/ml and 10 μl were added per well. Each condition was performed in triplicates. The plate was incubated in an automated plate reader under aerobic conditions at 37˚C. The $OD_{600}$ was measured every 60 min. Blank values were deducted manually and the average of three samples was calculated and plotted using Prism GraphPad8.2.

## Measurements of short-chain fatty acids

Approximately 50–100 mg of cecal content was collected, weighted and immediately snap-frozen in liquid nitrogen and stored at—80˚C until further processing. For extraction of SCFAs, samples were resuspended in 600 μl water spiked with internal standard (2 μl o-cresol/250 ml) and 60 μl 65% HPLC-grade sulfuric acid per 50 mg fresh weight and mixed vigorously for 5 min. Next, 400 μl of the mixture were extracted with 200 μl of tert-butyl methyl ether, and the ether phase was analyzed by GC-MS as described previously [40]. Standard curves of organic acids were used for external calibration.

## Metabolite derivatization

Online metabolite derivatization was performed using an Axel Semrau Autosampler. Dried polar metabolites were dissolved in 15 μL of 2% methoxyamine hydrochloride in pyridine at 40˚C under shaking. After 90 min, an equal volume of N-methyl-N-(trimethylsilyl)-trifluoracetamide (MSTFA) was added and held for 30 min at 40˚C.

## GC-MS analysis untargeted metabolome

Sample (1 μL) was injected into an SSL injector at 270˚C in split mode (1:5). GC-MS analysis was performed using an Agilent 7890A GC equipped with a 30m VF-35MS + 5m Duraguard capillary column (0.25 mm inner diameter, 0.25 μm film thickness). Helium was used as carrier gas at a flow rate of 1.0 mL/min. The GC oven temperature was held at 80˚C for 6 min and increased to 300˚C at a rate of 6˚C/min and held at that temperature for 10 min. Subsequently, the temperature was increased to 325˚C at a rate of 10˚C/min and held at that temperature for 4 min, resulting in a total run time of 60 min per sample. The GC was connected to an Agilent 5975C MS operating under electron impact ionization at 70 eV. The transfer line temperature was set to 280˚C. The MS source was held at 230˚C and the quadrupole at 150˚C. The detector was operated in scan mode. Full scan mass spectra were acquired from *m/z* 70 to *m/z* 800 at a scan rate of 2 scans/s. Tuning and maintenance of the GC-MS was done according to the supplier´s instructions, an automated tuning routine was applied every 150 injections. Data processing was done using the MetaboliteDetector software [41].

## DNA isolation

Feces samples were collected and stored at– 20˚C until processing for DNA based 16S rRNA gene sequencing. DNA was extracted using a phenol-chloroform-based method previously described [42]. In brief, 500 μl of extraction buffer (200 mM Tris (Roth), 20 mM EDTA (Roth), 200 mM NaCl (Roth), pH 8.0), 200 μl of 20% SDS (AppliChem), 500 μl of phenol: chloroform:isoamyl alcohol (PCI) (24:24:1) (Roth) and 100 μl of zirconia/silica beads (0.1 mm diameter) (Roth) were added per feces sample. Lysis of bacteria was performed by mechanical disruption using a Mini-BeadBeater-96 (BioSpec) for two times 2 min. After centrifugation, aqueous phase was passed for another phenol:chloroform:isoamyl alcohol extraction before precipitation of DNA using 500 μl isopropanol (J.T. Baker) and 0.1 volume of 3 M sodium acetate (Applichem). Samples were incubated at—20˚C for at least several hours or overnight and centrifuged at 4˚C at maximum speed for 20 min. Resulting DNA pellet was washed, dried using a speed vacuum and resuspended in TE Buffer (Applichem) with 100 μg/ml RNase I (Applichem). Crude DNA was column purified (BioBasic Inc.) to remove PCR inhibitors.

## 16S rRNA gene amplification and sequencing

16S rRNA gene amplification of the V4 region (F515/R806) was performed according to an established protocol previously described [43]. Briefly, DNA was normalized to 25 ng/μl and used for sequencing PCR with unique 12-base Golary barcodes incorporated via specific primers (obtained from Sigma). PCR was performed using Q5 polymerase (NewEnglandBiolabs) in triplicates for each sample, using PCR conditions of initial denaturation for 30 s at 98˚C, followed by 25 cycles (10 s at 98˚C, 20 s at 55˚C, and 20 s at 72˚C). After pooling and normalization to 10 nM, PCR amplicons were sequenced on an Illumina MiSeq platform via 250 bp paired-end sequencing (PE250). Using Usearch8.1 software package (http://www.drive5.com/usearch/) the resulting reads were assembled, filtered and clustered. Sequences were filtered for low quality reads and binned based on sample-specific barcodes using QIIME v1.8.0 [44]. Merging was performed using -fastq_mergepairs–with fastq_maxdiffs 30. Quality filtering was conducted with fastq_filter (-fastq_maxee 1), using a minimum read length of 250 bp and a minimum number of reads per sample = 1000. Reads were clustered into 97% ID OTUs by open-reference OTU picking and representative sequences were determined by use of UPARSE algorithm [45]. Abundance filtering (OTUs cluster > 0.5%) and taxonomic classification were performed using the RDP Classifier executed at 80% bootstrap confidence cut off

[46]. Sequences without matching reference dataset, were assembled as *de novo* using UCLUST. Phylogenetic relationships between OTUs were determined using FastTree to the PyNAST alignment [47]. Resulting OTU absolute abundance table and mapping file were used for statistical analyses and data visualization in the R statistical programming environment package phyloseq [48].

## Western blotting

Tissue was homogenized and cells were lysed and the protein content of the lysates was separated on a 10% SDS- polyacrylamide gel electrophoresis followed by Coomassie staining or transfer onto a PDVF membrane (Amersham / GE Healthcare Life Science). After blocking with 3% skimmed milk for 1 h at room temperature, membranes were incubated with the appropriate primary antibody overnight, followed by 1 h incubation with HRP-conjugated secondary antibodies at room temperature. HIF-1α and β-actin (loading control) were detected using rabbit anti-HIF-1α (Abcam 82832) and mouse anti- β-actin (Abcam 8226) at 1: 500 and 1:1000 respectively and an anti-rabbit or anti mouse secondary antibody using 1: 10000 dilution. Protein bands were visualized using ECL reagent (Advansta) using a western blot imaging system (BioRad) and quantified using Image J software.

## Supporting information

**S1 Fig. Isogenic mouse lines feature distinct microbiota compositions and do not lose significant body weight during *C. rodentium* infection.** Fecal microbiota of different specific pathogen free (SPF) mouse lines including SPF-1/SPF-S (n = 17), SPF-2/SPF-R (n = 18), SPF-3 (n = 7), SPF-4 (n = 9), SPF-5 (n = 15), SPF-6 (n = 4) and SPF-7 (n = 12) were evaluated using 16S rRNA gene sequencing. (A) α-diversity was determined using Chao1 and Shannon index. (B) Relative abundance at phylum level for each group is shown. (C) Ratio of taxonomic order *Firmicutes* and *Bacteroidetes* for each group. Representative data derived from one experiment (SPF-4, SPF-6) or are pooled from at least two different experiments (SPF-1, SPF-2, SPF-3, SPF-5, SPF-7). P values indicated represent a nonparametric Kruskal-Wallis test with multiple comparisons (one-way ANOVA). $^*p<0.05$, $^{**}p<0.01$, $^{***}p<0.001$, $^{****}p<0.0001$. (D) Relative abundances of bacterial families are shown and grouped according to their phylum. Bars represent the mean of all mice within the group. Representative data derived from one experiment (SPF-4, SPF-6) or are pooled from at least two different experiments (SPF-1, SPF-2, SPF-3, SPF-5, SPF-7). (E) Mice with different microbiota settings (SPF-1-SPF-7) were infected orally with $10^8$ CFU *C. rodentium*. Body weight was recorded during the course of infection. Results represent n = 4–18 mice/group as mean ± SEM from one experiment (SPF-4, SPF-6) or pooled from at least two different experiments (SPF-1, SPF-2, SPF-3, SPF-5, SPF-7).
(TIF)

**S2 Fig. Fecal microbiota strongly differs between SPF-S and SPF-R mice with higher ratios of SCFA producing bacteria in resistant mice SPF-R mice.** (A) Fecal microbiota of resistant SPF-R and susceptible SPF-S mice was analyzed using 16S rRNA gene sequencing using a principal coordinates analysis (PCoA) plot (B) α-diversity was determined using Chao1 and Shannon index. P values indicated represent a non-parametric Wilcoxon signed rank test $^{****}p<0.0001$. (C) Relative abundances are displayed at genus level. (D) Heatmap of relative abundances of bacterial genus ($\geq$ 97% sequence similarity, $> 0.05\%$ relative abundance) is sorted by genus and mouse line. Each vertical bar represents the microbiota of an individual mouse. Representative data are from one experiment out of three independent experiment. (E) Relative abundance of significantly different SCFA producing members between SPF-S and

SPF-R of the genus *Bacteroides*, *Intestinimonas*, *Lachnoclostridium*, *Roseburia* and *Ruminoclostridium* are shown. P values indicated represent a non-parametric Wilcoxon signed rank test *p<0.05, **p<0.01, ***p<0.001, ****p<0.0001. (F) pH value of the cecum at steady state in susceptible SPF-S and resistant SPF-R mice. *P* values indicated represent a Mann-Whitney U test comparison between groups with *p<0.05.
(TIF)

**S3 Fig. Cohousing experiments lead to various outcome in the phenotype with high abundance of SCFA producing bacteria in resistant mice.** (A) SPF-S and SPF-R mice were cohoused for 4 weeks and infected with $10^8$ CFU *C. rodentium*. (B) Assignment of cohoused SPF-S and SPF-R mice to the class "resistant" and "susceptible" according to the cecal CFUs of *C. rodentium* after day 3 p.i.. A threshold of $10^6$ was used for discrimination of both groups. (C) Fecal microbiota was analyzed using 16S rRNA gene sequencing after cohousing using a principal coordinates analysis (PCoA) plot. (D) α-diversity was determined using Chao1 and Shannon index. (E) Fecal microbiota of resistant and susceptible cohoused SPF-S and SPF-R mice was analyzed using 16S rRNA gene sequencing. Relative abundances of bacterial families are shown and grouped according to their phylum. Bars represent the mean of all mice within the group. Representative data derived from three independent experiment are pooled. (F) Relative abundance of significantly different SCFA producing members between resistant and susceptible mice of the genus *Alistipes*, *Butyricicoccus*, *Intestinimonas*, *Lachnoclostridium*, *Lachnospiraceae*, *Roseburia*, *Ruminococcaceae* and *Ruminoclostridium* are shown. P values indicated represent a non-parametric Wilcoxon signed rank test *p<0.05, **p<0.01, ***p<0.001, ****p<0.0001. (G) Relative levels of *Akkermansia* in SPF-S and SPF-R animals before and after three weeks of cohousing. Values are normalized to total 16S. (H) Relative levels of *Akkermansia* cohoused SPF-S and SPF-R mice that changed or maintained the initial phenotype. Values are normalized to total 16S. P values indicated represent a non-parametric Wilcoxon signed rank test *p<0.05
(TIF)

**S4 Fig. Isolated facultative anaerobic bacterial species do not contribute to inhibition of *C. rodentium in vivo*.** (A) HIF-1 α protein from cecum tissue at steady state in SPF-R and SPF-S mice (B) Quantification of HIF-1 α protein compared to β-actin protein levels (loading control) (C) Serial dilutions of isolated cecal content and feces of SPF-S and SPF-R mice were plated on MacConkey agar plates without crystal violet and cultivated for 24 hours under aerobic conditions. Morphological different colonies were picked and streaked out again to obtain a pure culture. Single colonies were sent for sequencing. All aerobic colonies were counted for quantification (D) Isolated bacteria grown on Mac Conkey agar plates. (E) Quantification of cultivable aerobic bacteria in the feces of SPF-1 and SPF-2 mice growing on Mac Conkey agar plates media. Data represent two independent experiments with n = 5–8 mice per group. (F) Groups of susceptible SPF-S mice were pretreated with isolated facultative anaerobic bacteria. All mice were infected with $10^8$ CFU *C. rodentium* after 3 weeks of precolonization and CFUs/g organ content and tissue were assessed after 3 days p.i. (G-H) CFUs of *C. rodentium* in intestinal organ tissues and contents after 3 days p.i.. Results represent Mean and SEM of two independent experiments with n = 6–10 mice per group. P values indicated represent a nonparametric Kruskal-Wallis test *p<0.05, **p<0.01, ***p<0.001, ****p<0.0001.
(TIF)

**S5 Fig. Untargeted metabolomics data of SPF-S and SPF-R mice before and after infection reveal strong differences between isogenic mouse lines.** Heat maps of successfully annotated and significantly different metabolites between SPF-S and SPF-R mice at different time points:

steady state (A) and day 1 p.i. (B). The two dendrograms for the heat map were calculated using Euclidean distance and ward linkage. *t*-tests with *p<0.05 were performed pairwise and significantly different metabolites between two extraction solvents are indicated. Fold changes of successfully annotated and significantly different metabolites between SPF-S and SPF-R at steady state (C) or at day 1 p.i. (D) are displayed. Fold changes of successfully annotated and significantly different metabolites within a mouse lines at different time points: SPF-S (E) and SPF-R (F). More than 2-fold changes between the indicated groups were set as a threshold.
(TIF)

**S6 Fig. The metabolomic profile of resistant SPF-R is characterized by elevated SCFA levels, which strongly impair the growth of *C. rodentium in vitro* at pH 6.0.** (A) *C. rodentium* growth displayed as optical density (OD) after 24 hours at pH 6.0 in BHI medium supplemented with different concentrations of acetate and propionate or without any SCFA added. (B) *C. rodentium* growth displayed as optical density (OD) after 24 hours at pH 6.0 (left) or pH (7.0) in BHI medium supplemented with different concentrations of acetate, butyrate and propionate or without any SCFA added. One data point represents mean value of three replicates. Values out of three independent experiments are displayed. *P* values indicated represent a one-way ANOVA between groups with **p<0.01, ***p<0.001, ****p<0.0001. (C-D) *C. rodentium* growth displayed as optical density over time at pH 6.0 (left) or pH (7.0) in BHI medium supplemented with different concentrations of acetate, butyrate and propionate or without any SCFA added. OD was measured every 60 min. Mean values ± SEM out of three independent experiments are displayed.
(TIF)

**S7 Fig. Short-term butyrate supplementation did not lead to major differences in the microbiome of SPF-S mice.** (A) Fecal bacterial microbiota composition of butyrate supplemented animals were evaluated using 16S rRNA gene sequencing regarding to their response against *C. rodentium* infection. β-diversity was analyzed using Bray-Curtis dissimilarity matrix and non-metric multidimensional scaling (NMDS). (B) α-diversity before and after SCFA supplementation in different groups of SPF-S mice was determined using Chao1 and Shannon index p>0.05. (C) Average microbiome level from different SPF-S mice on family level. (D-E) Cecal butyrate level and cecal pH of C57BL6/N germ-free mice supplemented with 150 mM in the drinking water at 3 days p.i. (F-G) Cecal butyrate level and cecal pH of antibiotic treated SPF-S mice supplemented with 150 mM in the drinking water at 3 days p.i. Mean and SEM of two independent experiments with n = 5–9 mice per group. P values indicated represent a nonparametric Kruskal-Wallis test *p<0.05, **p<0.01, ***p<0.001, ****p<0.0001
(TIF)

**S1 Table. Origin of isogenic mouse lines.**
(XLSX)

**S2 Table. Isolated facultative anaerobic bacteria from SPF-R cecal content.**
(XLSX)

## Acknowledgments

We thank the staff of the animal facility and the "genome analytics core facility" of the HZI for technical support as well as G. Martens for assistance with metabolomics data processing. We are grateful to Dr. M. Lochner for providing the bioluminescent *C. rodentium* strain and Dr. J. Leong for providing the Shiga toxin-producing *C. rodentium* strain.

## Author Contributions

**Conceptualization:** Lisa Osbelt, Sophie Thiemann, Till Strowig.

**Data curation:** Lisa Osbelt, Eric J. C. Gálvez.

**Formal analysis:** Lisa Osbelt, Sophie Thiemann, Till Robin Lesker, Kerstin Schmidt-Hohagen, Marina C. Pils, Meina Neumann-Schaal.

**Funding acquisition:** Dirk Schlüter, Till Strowig.

**Investigation:** Lisa Osbelt, Sophie Thiemann, Nathiana Smit, Madita Schröter, Kerstin Schmidt-Hohagen, Marina C. Pils, Meina Neumann-Schaal.

**Methodology:** Lisa Osbelt, Sophie Thiemann, Nathiana Smit, Till Robin Lesker, Eric J. C. Gálvez, Kerstin Schmidt-Hohagen, Marina C. Pils, Sabrina Mühlen, Petra Dersch, Karsten Hiller, Meina Neumann-Schaal.

**Project administration:** Till Strowig.

**Resources:** Lisa Osbelt, Sabrina Mühlen, Petra Dersch, Karsten Hiller, Till Strowig.

**Supervision:** Petra Dersch, Dirk Schlüter, Till Strowig.

**Visualization:** Lisa Osbelt, Sophie Thiemann.

**Writing – original draft:** Lisa Osbelt, Sophie Thiemann.

**Writing – review & editing:** Lisa Osbelt, Nathiana Smit, Kerstin Schmidt-Hohagen, Sabrina Mühlen, Petra Dersch, Karsten Hiller, Dirk Schlüter, Meina Neumann-Schaal.

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
