## [Decision Letter · Decision Letter 0]

4 Dec 2019

Dear Prof. Strowig,

Thank you very much for submitting your manuscript "Variations in microbiota composition of laboratory mice influence Citrobacter rodentium infection via variable short-chain fatty acid production" (PPATHOGENS-D-19-01806) for review by PLOS Pathogens. Your manuscript was fully evaluated at the editorial level and by independent peer reviewers. The reviewers appreciated the attention to an important problem, but raised some substantial concerns about the manuscript as it currently stands. These issues must be addressed before we would be willing to consider a revised version of your study. We cannot, of course, promise publication at that time.

We therefore ask you to modify the manuscript according to the review recommendations before we can consider your manuscript for acceptance. Your revisions should address the specific points made by each reviewer.

(1) A letter containing a detailed list of your responses to the review comments and a description of the changes you have made in the manuscript. Please note while forming your response, if your article is accepted, you may have the opportunity to make the peer review history publicly available. The record will include editor decision letters (with reviews) and your responses to reviewer comments. If eligible, we will contact you to opt in or out.

(2) Two versions of the manuscript: one with either highlights or tracked changes denoting where the text has been changed; the other a clean version (uploaded as the manuscript file).

Additionally, to enhance the reproducibility of your results, PLOS recommends that you deposit your laboratory protocols in protocols.io, where a protocol can be assigned its own identifier (DOI) such that it can be cited independently in the future. For instructions see http://journals.plos.org/plospathogens/s/submission-guidelines#loc-materials-and-methods

We hope to receive your revised manuscript within 60 days. If you anticipate any delay in its return, we ask that you let us know the expected resubmission date by replying to this email. Revised manuscripts received beyond 60 days may require evaluation and peer review similar to that applied to newly submitted manuscripts.

[LINK]

Sincerely,

Andreas J Baumler

Associate Editor

PLOS Pathogens

Nina Salama

Section Editor

PLOS Pathogens

Kasturi Haldar

Editor-in-Chief

PLOS Pathogens

orcid.org/0000-0001-5065-158X

Grant McFadden

Editor-in-Chief

PLOS Pathogens

orcid.org/0000-0002-2556-3526

Reviewer's Responses to Questions

**Part I - Summary**

Reviewer #1: In this study, the authors investigate the role of the microbiota in disease development and colonization of the mouse pathogen Citrobacter rodentium. The authors take advantage of isogenic mouse lines that harbor distinct microbiota communities in order to determine whether different microbiota compositions correlate with the pathogenesis of the organism. They found that variable colonization of C. rodentium in mice correlates with specific microbiota communities in mice, which they could recapitulate these phenotypes by co-housing using germ-free mice or susceptible mice with resistant or susceptible mice. They perform deeper analysis and find that there is a higher abundance of butyrate-producing microbiota and higher butyrate concentrations in “resistant mice.” Finally, the authors demonstrate that giving susceptible mice SCFAs reduces C. rodentium colonization, which they attribute to SCFA inhibition by performing in vitro incubation of C. rodentium with SCFAs. The authors conclude that varying concentrations of SCFAs, particularly butyrate, levels correlate the C. rodentium colonization in different mice by a direct SCFA inhibitory mechanism.

The experiments are well-performed and the data are interesting. My major criticism is that the mechanism of in vivo inhibition of C. rodentium colonization is not clear. The authors perform in vitro assays to demonstrate that addition of SCFA inhibits growth of C. rodentium. However, as the authors state in the discussion, the mechanisms of SCFA pathogen inhibition are complex and also include crosstalk with immune cells and modulation of host metabolism. Furthermore, resistant (SPF-2) mice harbored significantly more Enterobacteriaceae, which has also been shown the inhibit pathogen colonization. Whether other confounding factors are involved in their phenotypes (particularly in Fig. 7) is not clear since the authors do not perform any experiments to exclude these other possible mechanisms in their model.

Reviewer #2: In this study, Osbelt et al investigate how differences in microbiota composition affect the susceptibility and kinetics of Citrobacter rodentium infection in mice. Initially, the authors compare shedding of Citrobacter in the feces of closely related mouse lines that harbor different gut microbial communities, with various mouse lines exhibiting different shedding kinetics. Transfer experiments into gnotobiotic mice and co-housing experiments suggest that these effects are primarily due to differences in gut microbial communities, and not host genetics. In a challenge model with Shiga-toxin producing Citrobacter, differences in microbial communities translated into altered susceptibility of infection, including mortality and pathogen burden at various body sites. To identify a mechanism by which gut microbial communities influence infection kinetics, the authors make use of the fact that protection is only conferred to a subset of animals. Protection was associated with increased community richness, increased abundance of SCFA-producing Firmicutes, and increased levels of SCFA, in particular butyrate. In vitro, Citrobacter growth was inhibited by SCFA in a dose- and pH-dependent manner. Supplementation of the mouse drinking water with short chain fatty acids partially conferred protection to Citrobacter infection.

The overall topic of how the gut microbiota confers colonization should be of interest to a broad audience. The experiments are well-controlled and the conclusions justified by the data. The overall conclusion of the study fits well within our current framework; key findings relating to the mechanism, i.e. susceptibility of Enterobacteriaceae to SCFA, have been reported in similar form by other groups, e.g. the Monack lab and the Pamer lab.

**Part II – Major Issues: Key Experiments Required for Acceptance**

Reviewer #1: 1.) In Fig. 7C, the authors supplemented susceptible SPF-1 mice with a SCFA mixture consisting of butyrate, acetate and propionate in the drinking water, which correlated with lower CFU/g of C. rodentium in Fig. 7D. The authors point out that in five mice that achieved the “threshold” of 50 nmol/mg butyrate in the cecum, and lower CFU/g in the cecum (Fig. 7G), which recapitulated resistant SPF-2 mice. The authors conclude that, “…higher abundance and diversity of SCFA-producing bacteria in the cecum of SPF-2 mice lead to elevated butyrate levels and reduced pH value in the lumen of the cecum, thereby efficiently reducing growth of C. rodentium…” However, since the experiments in Fig. 7 were performed in the presence of conventional mice with a complex microbiota, it’s difficult to draw conclusions about the effects of SCFAs in these mice with such pronounced microbiota differences.

In Fig. 7G, the authors show that 5/19 mice achieved the “threshold” of 50 nmol/mg butyrate and lower CFU/g of C. rodentium in the cecum. However, it’s unknown whether these five mice were microbiota “outliers” with native higher levels of SCFAs independent of oral SCFA supplementation. Did the authors analyze the microbiota of these mice to examine whether the harbored higher levels of butyrate-producing microbiota or other inhibitory microbiota such as Enterobacteriaceae? Although the former observation would support their conclusions, it’s necessary to repeat these experiments in Germ-Free or antibiotic-treated mice. The authors should repeat the experiments in Fig. 7 in germ-free (or antibiotic treated mice) and show that SCFA supplementation alone is sufficient to reduce C. rodentium colonization to support their conclusion that SCFAs are responsible for these effects. The authors could also give germ-free mice butyrate-producing Firmicutes and measure SCFA, pH, and CFU/g C. rodentium in these mice.

Reviewer #2: Decreases in butyrate levels have been linked to increased epithelial oxygenation (Colgan group, Kelly et al., 2015 Cell Host and Microbe), and increased oxygenation enhances mucosal Citrobacter colonization (Baumler group, Lopez et al., 2016 Science). As it stands, the mechanism proposed by authors is plausible, but this alternative hypothesis cannot be formally excluded and could provide an alternative explanation of all the data. Additional experimentation is recommended to discern between these two possibilities.

**Part III – Minor Issues: Editorial and Data Presentation Modifications**

Reviewer #1: 1.) Since SCFA’s supplemented orally are readily absorbed in the small intestine, this may explain the significantly lower butyrate concentrations in Fig. 7E and CFU/g of SPF-1 + SCFA mice in Fig. 7D, relative to SPF-2 mice. Oral treatment of mice with tributyrin has been shown to increase concentrations of butyrate in the large intestine. The authors could give mice (and germ-free or antibiotic treated) tributyrin instead of free SCFA to see if this could restore butyrate levels and reduce colonization in mice.

2.) It would be helpful for the reader if the authors changed the names of the different mice from SPF-1 and SPF-2 to SPF-S and SPF-R, respectively, to make it easier to follow with mice are resistant and which are susceptible.

Reviewer #2: Data presentation is very clear and concise.

PLOS authors have the option to publish the peer review history of their article (what does this mean?). If published, this will include your full peer review and any attached files.

Reviewer #1: Yes: Fabian Rivera-Chavez

Reviewer #2: No

---

## [Editor Report · Decision Letter 1]

1 Mar 2020

Dear Prof. Strowig,

We are pleased to inform you that your manuscript 'Variations in microbiota composition of laboratory mice influence Citrobacter rodentium infection via variable short-chain fatty acid production' has been provisionally accepted for publication in PLOS Pathogens.

Best regards,

Andreas J Baumler

Associate Editor

PLOS Pathogens

Nina Salama

Section Editor

PLOS Pathogens

Kasturi Haldar

Editor-in-Chief

PLOS Pathogens

orcid.org/0000-0001-5065-158X

Michael Malim

Editor-in-Chief

PLOS Pathogens

orcid.org/0000-0002-7699-2064
---

## [Editor Report · Acceptance letter]

17 Mar 2020

Dear Prof. Strowig,

We are delighted to inform you that your manuscript, "Variations in microbiota composition of laboratory mice influence *Citrobacter rodentium* infection via variable short-chain fatty acid production," has been formally accepted for publication in PLOS Pathogens.

Best regards,

Kasturi Haldar

Editor-in-Chief

PLOS Pathogens

orcid.org/0000-0001-5065-158X

Michael Malim

Editor-in-Chief

PLOS Pathogens

orcid.org/0000-0002-7699-2064